# Space-environment relationship in the identification of potential areas of expansion of *Trypanosoma cruzi* infection in *Didelphis aurita* in the Atlantic Rainforest

Raphael Testai[1,2], Marinez Ferreira de Siqueira[3], Diogo Souza Bezerra Rocha[4], Andre Luiz Rodrigues Roque[1], Ana Maria Jansen[1], Samanta Cristina das Chagas Xavier[1]*

1 Laboratory of Tripanosomatid Biology, Oswaldo Cruz Institute, Oswaldo Cruz Foundation (FIOCRUZ), Rio de Janeiro, Brazil, 2 The Graduate Program in Computational and Systems Biology of the Instituto Oswaldo Cruz (PGBCS/IOC/Fiocruz), Rio de Janeiro/RJ, Brazil, 3 Research Institute of the Botanical Garden of Rio de Janeiro, JBRJ, Rio de Janeiro/RJ, Brazil, 4 International Institute for Sustainability, IIS, Rio de Janeiro/RJ, Brazil

* samanta@ioc.fiocruz.br

**Data Availability Statement:** DOI from the GBIF (2019) occurrence download for T. vitticeps: https://doi.org/10.15468/dl.qr4ykt DOI from the

## Abstract

Ecological Niche Modeling is widely used for animals, but rarely for understanding the parasite ecology. *Trypanosoma cruzi* is a heterogeneous and widely dispersed multi-host parasite. *Didelphis aurita* is a generalist species, both in terms of diet and environments. We modeled the *D. aurita* niche and *T. cruzi* infection in the Brazilian Atlantic Rainforest, using the models of two common vector species (*Triatoma vitticeps* and *Panstrongylus megistus*) as biotic variables, predicting their occurrence. Records of *T. cruzi* infected and non-infected *D. aurita* were analyzed through climate and landscape approaches by the Ecoland method. Models for each triatomine species and infected and noninfected *D. aurita* were produced considering climate and landscape: resolution of ~1km² selected by Pearson's correlation [-0.7≤α≤0.7]. For modeling, seven algorithms available in ModleR package were used. True Skill Statistic was used to evaluate the models' performance (≥ 0.7). *T. vitticeps* indicates that there is a spatial dependence with warm areas in the southeastern region while *P. megistus* presented a distribution with high environmental suitability concentrated in the Southeast. High values of climatic suitability, landscape and potential presence of *T. vitticeps* and *P. megistus* were considered necessary, but not sufficient for the presence of *D. aurita* infected by *T. cruzi*. Climate models showed an ecological niche with suitability variations homogeneous, and landscape models showed a distribution of habitat conditions along the biome, with a fragmented profile and heterogeneous between locations. Ecoland demonstrated that *D. aurita* has different degrees of impact on its role in the enzootic cycle in different locations of the Atlantic Rainforest. Associating the models with the Ecoland method allowed the recognition of areas where *D. aurita* are important *T. cruzi* reservoirs. Areas of high suitability for the presence of marsupials are a necessary, but not sufficient for *D. aurita* to act as a reservoir for *T. cruzi*.

GBIF (2020) occurrence download for P. megistus.: https://doi.org/10.15468/dl.b7ak27 Search code from the Specieslink.net (2020a) occurrence download for D. aurita: 4846_20201102093759 Search code from the Specieslink.net (2020b) occurrence download for T. vitticeps: 23695_20190401123424 Search code from the Specieslink.net (2020c) occurrence download for P. megistus: 20774_20190408111620 Search code from the Specieslink.net (2020d) occurrence download for P. megistus: 20886_20190408111650 The availability of the Trypanosomatid Biology Laboratory (LABTRIP) database for D. aurita, T. vitticeps and P. megistus should occur by contacting: samanta@ioc.fiocruz.br. and/or labtrip@ioc.fiocruz.br.

**Funding:** Funding: this study was funded by Fundação Oswaldo Cruz (Fiocruz), Conselho Nacional de Desenvolvimento Científico e Tecnológico (CNPq), Fundação Carlos Chagas Filho de Amparo à Pesquisa do Estado do Rio de Janeiro (FAPERJ), Coordenação de Aperfeiçoamento de Pessoal de Nível Superior (CAPES). AR is financially supported by CNPq/Universal (425293/ 2018-1) and Jovem Cientistas do Nosso Estado/ Faperj (E-26/202.794/2019). AJ is financially supported by CNPq (Bolsista de Produtividade, nível 1A). SX has received financial support from CNPq (MCTIC/CNPq No. 28/2018 - Universal, process number 422489/2018-2), and JCNE/ FAPERJ (E-26/201.314/2021), AR and SX Faperj (Apoio a Grupos Emergentes de Pesquisa no estado do Rio de Janeiro, process number E-26/ 010.002276/2019). The funders had no role in study design, data collection and analysis, decision to publish, or preparation of the manuscript.

**Competing interests:** The authors declare they have no conflict of interest.

## Introduction

The protozoan *Trypanosoma cruzi* (Kinetoplastida: Trypanosomatidae) is the etiologic agent of Chagas disease (CD), with wide distribution in nature, occurring from the southern United States to southern Argentina and Chile. It is a parasite immersed in complex trophic nets, which infects hundreds of domestic and wild mammals species [1–3]. The transmission involves hematophagous insects of the subfamily Triatominae (Hemiptera: Reduviidae) [4] divided into five tribes, 18 genera and 154 species [5], while *T. cruzi* is a heterogeneous parasite, in which six genotypes or Discrete Typing Units (DTUs) are recognized, TcI to TcVI, besides a seventh DTU known as Tcbat [6–8].

*T. cruzi* trypanosomiasis is primarily a sylvatic, enzootic, in which each animal species has a different role in the trophic network, in space and time, in terms of its maintenance and infective competence [9], resulting in distinct enzootic scenarios. That is, each region is peculiar, presents a specific transmission network, and its understanding is necessary to define the risk areas for human disease [10].

In Brazil, it is recorded 68 triatomine species, and 13 species from three genera (*Panstrongylus*, *Triatoma* and *Rhodnius*) are recognized of epidemiological importance due to behavioral characteristics, some of them presenting wide spatial distributions [11,12]. Two of these species in the Atlantic Rainforest are: (i). *Panstrongylus megistus*—central in the biome, being one of the species with the widest distribution in the country, occurring in more than 20 of the 26 states and three different biomes, reflecting the great capacity of these vectors to adapt to different environmental conditions; and (ii) *Triatoma vitticeps*, being very frequent in rural areas with high prevalence in the state of Espírito Santo, but also present in the states of Rio de Janeiro, Minas Gerais and Bahia [13–15]. *P. megistus* is often found in hollow trees in arboreal environments, which are also inhabited by *Didelphis* spp. [16], while *T. vitticeps* is capable of forming colonies associated with marsupial nests in peridomicile areas [14,15].

The infective potential of a given host to transmit a hematozoan, as is the case of *T. cruzi*, depends on the concentration of parasites present in the bloodstream (parasitemia), which is a variable that changes as a function of time, host characteristics, genetic background of the parasite, health status, among others [1,17]. To verify the parasitemia of a particular species, parasitological tests are performed (*e.g.* hemoculture, blood fresh examination and *xenodiagnosis*), which aim to detect viable parasites in the bloodstream, and the positive result in these tests indicates the infective potential of that host [1,17].

The Atlantic Rainforest biome presents enzootic cycles infecting species of the orders Carnivora, Chiroptera, Didelphimorphia, Primates and Rodentia, with a higher prevalence of TcI and TcII *T. cruzi* DTUs [1]. The species *Didelphis aurita* and *Philander frenatus*, both from the order Didelphimorphia, are the species that present the highest *T. cruzi* infection rate, detected by positive blood culture, in this biome [1]. Actually, *D. aurita* is considered an important *T. cruzi* reservoir and a bioaccumulator of trypanosomatids, being able to maintain high and long-term parasitemias [17,18] with different *Trypanosoma* species and *T. cruzi* DTUs. This is a factor that, added to the increase in anthropization and the intensification of climate change, and the fact that this is a generalist species that inhabit anthropic areas, makes *D. aurita* a key species in the spatial analysis of *T. cruzi* distribution in the Atlantic Rainforest. The *Didelphis* genus is the one with the greatest dispersion in the American continent, from southeastern Canada to southern Argentina, being highly adaptable to different ecological niches, inhabiting mainly regions with high degrees of anthropic action. Furthermore, this genus is considered an important *T. cruzi* reservoir [19].

A possible way to analyze the relationship of *D. aurita* in the transmission cycle of *T. cruzi* in the Atlantic Rainforest is through spatial analysis. This is a tool of fundamental importance

for ecoepidemiological studies, which has been increasingly used in studies of vector-borne parasites, especially involving multi-host parasites, as is the case of Chagas disease, schistosomiasis and American visceral leishmaniasis [20–22], identifying areas at risk of parasitic transmission and the interactions among parasites, hosts and the environment [21,22]. In this context, Ecological Niche Modeling (ENM) can be applied to explore the existing relationships in the host/parasite/environment triad, in this case, the host *D. aurita* and its parasite *T. cruzi*. The ENM can estimate the Existing Fundamental Niche ("potential niche") of a species through the environmental characteristics where its points of occurrence are located, resulting in models that represent areas with adequate environmental characteristics for its maintenance [23,24]. Occurrence data and environmental variables are processed in Machine Learning algorithms, estimating areas with favorable environmental conditions for their occurrence [25–27]. An interesting way to simplify the analysis of ecological niche models is using the Ecoland approach, developed by Ferro e Silva et. al. [28]. As the models generated by climate and landscape variables present different response patterns, this approach performs a comparative analysis by applying map algebra between the models generated in a climate and landscape approach, separately. This allows the qualitatively and quantitatively classification of the models in terms of high, medium and low suitability (β) of agreement and disagreement between these approaches, mapping suitable areas for the presence of a species in a simplified way [28].

Therefore, the following questions arise: what are the conditions related to the triad parasite/host/environment in which the transmission of *T. cruzi* involving *D. aurita* occurs? What are the biotic and abiotic conditions in which this maintenance of the cycle takes place? We can hypothesize that the spatial distribution of triatomines, *D. aurita* and its infection by *T. cruzi* is directly related to the physiognomy of the landscape; and that ENM and the Ecoland approach are sensitive tools for detecting those areas. In this study, we applied the ENM for the interaction between *D. aurita* and *T. cruzi* in the Atlantic Rainforest biome, using both abiotic (environmental) and biotic (entomological) variables, as a basis for the distribution of two vector species: *T. vitticeps* and *P. megistus*. In this way, an environmental model was generated to support parasitology studies through the current distribution of this relationship between host/parasite/vector/environment, providing hotspot areas of *T. cruzi* infection in *D. aurita* and its distribution in the Atlantic Rainforest.

## Materials and methods

### Study design

Six ecological niche models were carried out: i) two triatomine models: *T. vitticeps* and *P. megistus*; ii) two models in the climate approach: *D. aurita* and *D. aurita* with positive blood culture; iii) and two models in the landscape approach: *D. aurita* and *D. aurita* with positive blood culture. Triatomine models *T. vitticeps* and *P. megistus* were introduced as explanatory variables in climate and landscape modeling of *D. aurita* with positive blood culture. As the climate and the landscape present different responses in relation to the niche and habitat of a species, after generating the 4 models of the marsupial, the Ecoland method [28] was applied to analyze the high, medium and low concordances by map algebra, climate and landscape suitability in the Brazilian Districts. In addition, the variables of *D. aurita* and *D. aurita* with positive blood cultures were analyzed using the Partition of Variance method, verifying the degree of their contribution to the models.

### Ethical approval and consent to participate

*D. aurita* were captured in accordance with the Normative Instruction of the Brazilian Institute of the Environment and Renewable Natural Resources (IBAMA n˚154/2007 Licenses

13373–1 and 19037–1), Chico Mendes Institute for Biodiversity and Conservation (ICMBIO, license number 13373), and Environmental Institute of Rio de Janeiro state (INEA, license number 020/2011). Procedures with animals were previously approved by the Animal Use Ethics Committee of the FIOCRUZ (CEUA/FIOCRUZ), licenses P0179-03, L0015-07, LW81-12, L-050-16.

## Study area

The Atlantic Rainforest biome is considered the second largest Brazilian biome, with one of the largest forest diversities per square meter in the world, with an area that originally covered 15% of the territory, equivalent to the 1,296,446 km$^2$ [29]. However, with the process of deforestation and anthropization, its area of coverage has become increasingly reduced over the years, and currently, only 12.4% of its original area is well conserved [30].

Approximately 270 mammal species, including 73 endemic species of mammals, were already reported for this biome. Its forest composition is formed by the dense rain forest, open rain forest, mixed rain forest, semideciduous seasonal forest, deciduous seasonal forest, savanna, steppe savanna, steppe, areas of pioneer formations and vegetation refuges [29]. Regarding the climate, the Atlantic Rainforest covers a large part of the Brazilian coast, receiving influences from the maritime humidity, which suffers impacts with the mountainous regions, promoting a high rate of rainfall in these places, reaching up to 4500 mm per year [31]. It presents a great variation of temperature and precipitation, something that, due to the presence of super humid equatorial regions without dry periods to temperate regions with dry winter and hot summer (Cwa), promotes different selective pressures to life forms [31,32].

## Database of occurrences

Data on the triatomines *T. vitticeps* and *P. megistus* and the marsupial *D. aurita* were collected through the databases of LABTRIP (Laboratory of Trypanosomatid Biology/IOC Fiocruz), Laboratory of Biology and Parasitology of Wild Mammals Reservoirs (IOC Fiocruz), Nucleus of Entomology and Malacology from the Espírito Santo State Health Department (Nemes—SESA/ES), Tropical Medicine Institute from the Espírito Santo Federal University and SpeciesLink. For triatomines, in addition to these data sources, the Global Biodiversity Information Facility–GBIF database was also included. The resulting triatomine database was composed by 721 occurrences of *T. vitticeps* (47.75%) and 789 of *P. megistus* (52.25%) from 1997 to 2019, comprising data from Specieslink (18.2%; n = 275), GBIF (42.12%; n = 636) and LABTRIP (39.68%; n = 599). The *D. aurita* database accounts for a total of 415 occurrence points, composed of data from the Laboratory of Biology and Parasitology of Wild Mammals Reservoirs, from Instituto Oswaldo Cruz (6.5%; n = 27), LABTRIP (56.4%; n = 234) and Specieslink (37.1%; n = 154), in which 80 (19.27%) specimens were infected by *T. cruzi*, identified through the positive result of the blood culture test, from 1992 to 2019. The geodesic reference system of the points of occurrence was the WGS-84 geocentric datum (EPSG:4326).

## Ecological Niche Modeling

For the occurrence of *T. vitticeps*, *P. megistus* and *D. aurita*, spatial filters were applied so that the models were generated with the least possible influence of difficult-to-measure variables and systematic errors, such as the bias of the models caused for the collection effort and its consequent influence on the densification of occurrences and environmental characteristics in each capture area. The following filters were applied to the databases: i) filter to remove points with duplicate coordinates; ii) filter for uniqueness of occurrence per pixel; iii) and filter to remove occurrences in pixels without data.

For the species modeling, presence-only algorithms and algorithms that use data of presence and absence of a species were used. In this case it was necessary to generate pseudo-absences for each of the species in places with no environmental suitability and/or low environmental suitability. For this, an inclusion buffer was considered (area Ψ, being a maximum limit for sampling the pseudo-absences) and an exclusion buffer (area Ω, region where the points were not generated because it is very close to the presences), so that the pseudo-absence points were generated in the difference area between them (Ψ – Ω). The application of this geographic filter to generate the pseudo-absence points in this area Ψ – Ω around each occurrence allows a significant improvement in the results [33]. The *T. vitticeps* exclusion buffer radius was 10 km [34], while its inclusion buffer was 60 km, approximately 90% of the median of the matrix of distances between occurrences. The exclusion buffer for *P. megistus* was the same as for *T. vitticeps*, while the inclusion buffer, due to its greater environmental diversity because of its greater spatial distribution along the Atlantic Forest, was defined as being 40% more than median of the matrix of distances between occurrences of *T. vitticeps*. For *D. aurita*, due to the fact that it is a nomadic species with great displacement capabilities and presence in anthropic areas [35,36], its exclusion buffer was defined as twice the buffer of *T. vitticeps* (20 km), while its inclusion buffer was 100 km, the same as *P. megistus*. The radii for each species can be seen in S1 Table.

Aiming to define a cut-off value for selecting areas of low environmental suitability to generate the pseudo-absences, the presence points of each species were analyzed using an environmental envelope technique, building a multidimensional bounding box of their environmental characteristics of minima and maximum occurrences of each species (Bioclim), defining a percentile, and using it to predict the environmental suitability of a given location. The environmental suitability of the presence points was verified in this generated model, so that the pixels with values in the range of > 90% suitability of the presences were considered as presence areas, while the pixels with values in the range of < 10% were considered areas of absence, which were defined as an interval to produce the pseudo-absences. The use of this environmental filter makes it possible to obtain an improvement in the results [33].

Subsequently, the databases were segregated into 5 partitions, in which 20% of the occurrences were used for testing and 80% for training the algorithm, applying the k-fold Cross-validation methodology, with two iterations.

The environmental variables were divided into climate and landscape. The 19 bioclimatic variables were acquired through the Worldclim website (https://worldclim.org/) in version 2.0, at 30 arc-second resolution (~ 1 km²), in raster format (GeoTIFF). The selection of climate and landscape variables was defined by Pearson's correlation between them, choosing those that have a correlation in the closed interval of $-0.7 \leq \alpha \leq 0.7$. Pearson's correlation was calculated considering only the pixels present in the Atlantic Forest area, increased by 50 km, considering the random selection of 1000 pixels. The landscape variables were obtained through the Google Earth Engine platform (GEE, https://earthengine.google.com/), which provides collections of images from orbital sensors, all in raster format (GeoTIFF). For the dataset of *D. aurita*, which has a temporal range from 1992 to 2019, the criterion that was defined that the temporal resolution of the GHG satellite images must encompass 80% of this data, acquiring images with a temporal resolution of 2007 until 2020. All environmental variables (climate and landscape) used for each approach are described in the reference listed in S2 Table.

The variable "Euclidean Distance to the Nearest Population Presence" (EDNPP) was produced from the image acquired from the GHG "GHSL: Global Human Settlement Layers, Population Grid 1975-1990-2000-2015 (P2016)" (S1 Appendix), which has a population estimate by resolution element (pixel) at 250 m resolution, which for Brazil is based on the IBGE demographic census. This variable was resampled to 30 arc-second (~ 1 km²), considering the

maximum pixel value within the final pixel area. Subsequently, the Euclidean distance from a pixel to the nearest pixel that has some population count was calculated. The variable "Euclidean Distance to the Nearest Drainage Section" (EDNDS) was calculated from the vector feature of the drainage sections of the IBGE geographic database on a scale of 1:1,000,000 (BCIM), from 2016, and from 1:250,000, 2019. Its calculation has the same methodology applied to the previous variable. The Normalized Difference Vegetation Index (NDVI) was generated using the MOD13A2 V6 product (S2 Appendix) by processing the bands of the near-infrared electromagnetic spectrum with the red (visible) of the images from the MODIS sensor of the Terra satellite, in the period from 01/01/2007 until 01/01/2020, at 926,625 m resolution. A single image was produced by calculating the median between the pixels of its NDVI image collection, which was later resampled to 30 arcseconds ($\sim 1$ km$^2$) by the nearest neighbor method (S2 Table).

Geographic filters allow the reduction of the bias of the points of presence in the models, considering that they are removing points close to each other that have redundant environmental characteristics, improving the results of the models [37]. The spacing was performed considering that the occurrence points must be 1.5 km apart (50% more in relation to the spatial resolution).

However, as *D. aurita* was modeled using NDVI in a landscape approach, to perform this redundancy correction of the environmental characteristics, the removal of nearby points was carried out associating them with this variable, thus avoiding the loss of important information of the vegetation for the species, as the pixel values of this index can vary abruptly between neighboring pixels. For this, the occurrence points that were in intersection areas within the same radius of 1.5 km from each point would be removed according to the NDVI value of these points, removing all that were below 0.2, and/or keeping the point with the highest NDVI value in that area, assuming it is greater than 0.2. This is because approximately 99% of the occurrence points of the *D. aurita* bank are contained above this value. For *T. cruzi* infection, there was no need to apply this filter, as the number of points was already low (29 points), and the points were already spaced apart (>1.5 km).

In relation to the triatomines *T. vitticeps* and *P. megistus*, despite the proximity between the occurrence points, the geographic filter was not applied due to the need to observe the suitability with the highest densities of occurrences. Furthermore, for both triatomines, the NDVI was modeled together with the climatic variables, thus avoiding the loss of important vegetation information for the species due to geographic spacing [37].

After modeling M1, M2, M3, M4 and M5 (S2 Table), the Partition of Variance [38–41] was carried out to verify the degree of contribution (Aj R$^2$) of each of the variables in the definition of the ecological niche for each of the Ensemble models generated, using the Venn Diagram to visualize the results. In the case of models generated by climate and landscape variables for *D. aurita* (M2 and M4), the components were divided into two groups of variables, temperature (BIO 2, BIO 4, BIO 5, BIO 8 and BIO 9) and of precipitation (BIO 12, BIO 13 and BIO 15), in the climate, while for the landscape each variable was presented as a component of the diagram (EDNHP, EDNDS and NDVI), that is, three components were generated. However, for the climate and landscape models for *D. aurita* infected by *T. cruzi* (M3 and M5), the Venn Diagrams were divided into 4 components in both cases, with the intention of allowing a better visualization and understanding of the temperature variables in the climate approach. For M3, two climatic components were produced (BIO 2, BIO 4 and BIO5) and (BIO 8 and BIO 9), a precipitation component (BIO 12, BIO 13 and BIO 15) and a component with the variable that represents the ecological niche models of triatomines (Triatomines). For M5, each variable was presented as a component of the diagram (EDNHP, EDNDS and NDVI), that is, three components were generated, and the fourth component was the variable that represents the

ecological niche models of triatomines (Triatomines). This analysis was performed with R package vegan 2.6–4.

**Platform and process flow of modeling.** The ModleR platform was used, implemented in the R programming language, and developed by the research team of the Jardim Botânico do Rio de Janeiro (JBRJ) [42], using 8 (eight) algorithms for processing, namely Bioclim, Maxent, Random Forest, SVM (Support Vector Machine, in two approaches), GLM (Generalized Linear Model), Mahalanobis Distance, DOMAIN and BRT (Boosted Regression Trees).

The evaluation of the models was carried out through the TSS (True Skill Statistics), being used as a cut-off threshold to define the selection of models with good results: TSS values $\geq 0.7$. Boxplots of the results obtained in the TSS of each of the algorithms used were defined, allowing the visualization and analysis of which of them best responded to the distribution of the species and the approach. For triatomines, only the algorithms' partitions that have uniformity of results among themselves were selected, with the criterion that the boxplots of the algorithms were not dispersed among themselves and that they have a minimum performance of TSS $\geq 0.7$. However, if the interquartile ranges were dispersed for all algorithms, partitions with TSS $\geq 0.7$ were selected, regardless of the algorithms. Furthermore, specifically for models of *D. aurita* infected or not by *T. cruzi*, as a criterion for selecting good quality partitions, partitions with TSS $\geq 0.7$ were selected—regardless of the agreement between algorithms—and with values of thresholds or cutoff thresholds >10%.

The Ensemble models for triatomines were generated through the weighted average between the TSS results of the partitions, while for the models of *D. aurita* with and without infection were generated through the average between the selected partitions (Fig 1). In the modeling of *D. aurita* with positive blood culture, the ecological niche models of *T. vitticeps* and *P. megistus* were inserted, both in the climate and landscape approach, to use them as biotic and explanatory variables for the presence of *T. cruzi* infection (Fig 1). The ENM, using landscape and climate variables, present different responses in each of these approaches. Thus, for *D. aurita*, four models were generated: two models considering marsupials infected by *T. cruzi*, in the climatic and landscape approaches; and two models for the species *D. aurita*, also in the climatic and landscape approaches.

## Comparative analysis between climate and landscape approaches

An analysis was performed comparing the climate and landscape models of *D. aurita* and its infection by *T. cruzi* through the Ecoland method [28], to verify the places of agreement and disagreement between them, indicating the regions with the greatest possibilities of environmental suitability for the occurrence of *T. cruzi* infection in the Atlantic Rainforest biome (Fig 1). For this purpose, the Districts (available on the IBGE website) were defined as the analysis scale, in the biome area, which were classified into three categories of environmental suitability ($\beta$), being high ($\beta \geq 66.67\%$), medium ($33.33\% \leq \beta < 66.67\%$) and low ($0 \leq \beta < 33.33\%$). The classification criterion was performed by the proportion of pixels of each class within the district area in relation to its total pixels. The class with the highest percentage of pixels within the district was the one that classified it. Thus, after classification, the value of its suitability was calculated by the average of the 50% highest values of this class.

For the Ecoland analysis for *T. cruzi*-infected *D. aurita*, models in climate and landscape approaches were compared to verify agreement and disagreement at each pixel in the Atlantic Rainforest extension. For another Ecoland analysis, of the *D. aurita* distribution, given that the triatomine *T. vitticeps* and *P. megistus* models were not included in the modeling, a sum (arithmetic operation) was performed between these two models to generate a single and resulting model of areas of potential suitability for both triatomines, and later this resulted model was

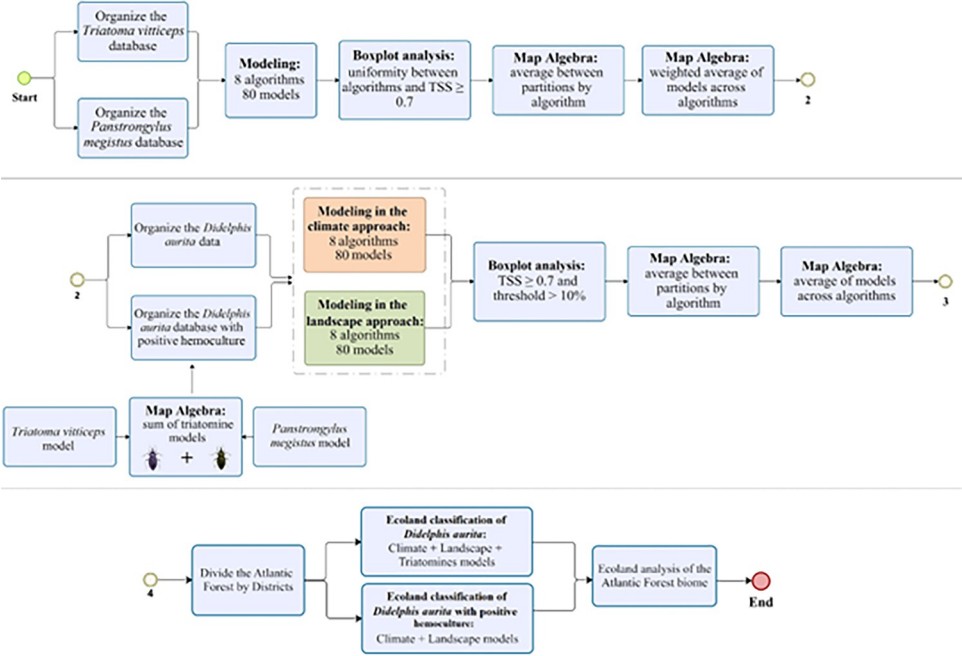

**Fig 1. Project process flow: 1) organization of the project Triatominae bank and its modeling (climate + landscape) of Triatomines.** Database, True Skill Statistics (TSS); 2) modeling climate and landscape approaches for *Didelphis aurita*, considering the result of hemoculture and without considering hemoculture; and 3) Analysis process flow by Ecoland. Positive hemoculture (h.c. +). Software: Bizagi Modeler. Software: Bizagi Modeler.

compared together with the marsupial climate and landscape models by map algebra. This allowed us to verify the areas of intersection between the presence of triatomines in relation to the models (climate and landscape) of *D. aurita*.

## Results

### Distribution of presence and pseudo-absence points of triatomines in the Atlantic Rainforest

After applying spatial filters in the triatomine databases, the value for *T. vitticeps* became 439/721 (60.9%) points, and for *P. megistus* 67/789 (8.49%) points. The pseudo-absences of *T. vitticeps* and *P. megistus* were generated, respectively, at values of environmental suitability < 2% for and < 1.9%, thus defining areas of low or no environmental suitability for both species (Fig 2).

### Distribution of the presence and pseudo-absence points of *Didelphis aurita* in the Atlantic Rainforest

After applying the spatial filters, the database for *D. aurita* was 168/415 (40.50%), and for its infection with *T. cruzi*, 27/80 (33.75%). The distribution of pseudo-absences for *D. aurita* in the climatic and landscape approaches were generated, respectively, in the suitability intervals < 0.78% and < 5.2%. For *D. aurita* infected by *T. cruzi*, in the climatic and landscape approaches, as the amount of presence points filtered in the space was low (27/80), there was no discrepancy in values between them, and 10% of these points are equivalent to only two occurrences, so that in both approaches the pseudo-absences were generated in areas without environmental suitability (equal to 0%) (Fig 3).

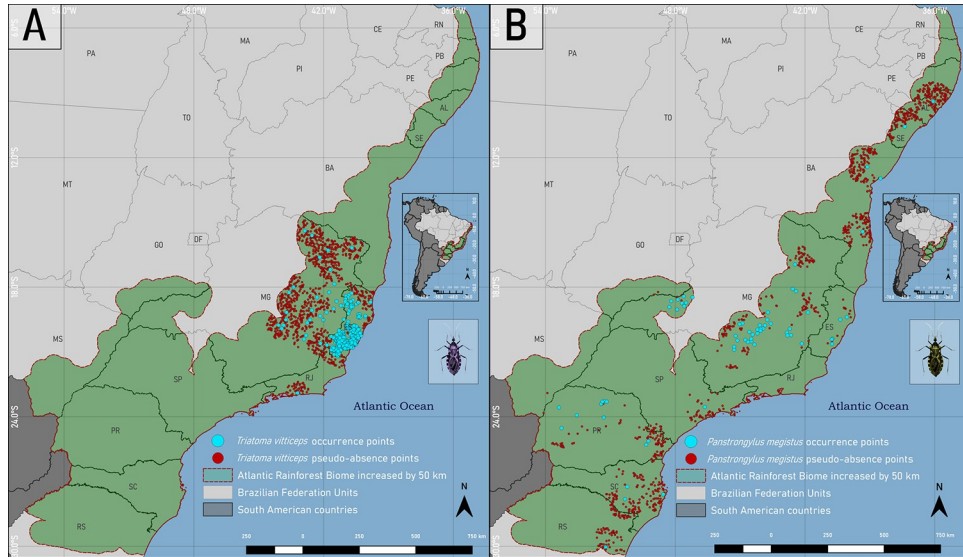

**Fig 2.** Distribution map of the occurrence and pseudo-absence points of *Triatoma vitticeps* (A) and *Panstrongylus megistus* (B) in the Atlantic Rainforest biome. Software: QGIS 3.22. Source: 1. Instituto Brasileiro de Geografia e Estatística (IBGE). Continuous cartographic base at 1:1,000,000 scale (2016). Available from: https://www.ibge.gov.br/geociencias/cartas-e-mapas/bases-cartograficas-continuas/15759-brasil.html?edicao=16033&t=downloads; Biomes and Coastal-Marine System of Brazil at 1:250,000 scale. Available from: https://www.ibge.gov.br/geociencias/cartas-e-mapas/informacoes-ambientais/15842-biomas.html?=&t=downloads.

## Ecological Niche Modeling of *Triatoma vitticeps* and *Panstrongylus megistus*

For the modeling of *T. vitticeps* (Fig 4), the algorithms that obtained the best performances and TSS boxplot intervals with overlapping were: SVM, Random Forests, Maxent and BRT, totaling 40 partitions, in such a way that the cut-off value for the TSS was the minimum value of this BRT statistic (TSS ≥ 0.86), being used to generate the final ecological niche model of the species in the Atlantic Rainforest. The Mahalanobis, GLM and Domain distances were below the cutoff value, so they were removed from the modeling.

For the modeling of *P. megistus* (Fig 4), the SVM, Maxent, Random Forests and BRT algorithms returned the best results, but with a greater variability compared to *T. vitticeps*, so that the TSS ≥ 0.7 was considered as a cutoff value, using them to generate the final model of *P. megistus*, through 43 partitions. The GLM and Domain algorithms, despite presenting TSS results above the cut-off, showed high TSS variability (0.50–0.87 and 0.48–0.88, respectively), below the cut-off, something that differs from the BRT because, despite also having a large TSS range in the boxplot, presented a smaller interquartile range of 0.0934. Regarding the Mahalanobis distance, it was the one that presented the most unsatisfactory results, in which, in addition to low TSS values of 0.15–0.57, it has the highest interquartile range of the model: 0.1695 (Fig 5 and Table 1).

## Ecological Niche Modeling of *Didelphis aurita* with and without *Trypanosoma cruzi* infection in climate and landscape approaches

For *D. aurita*, both in the climatic and in the landscape approach, partitions with a cut-off threshold above 10% and TSS ≥ 0.7 were selected. In the climatic one, all the algorithms were used to produce the Ensemble model of *D. aurita* (Fig 6A and S3 Table), composing it from 51 partitions. The SVM, Random Forest, Maxent and BRT algorithms returned the best results, ranging from 0.78–0.99, while the GLM and Domain obtained the lowest TSS values, from

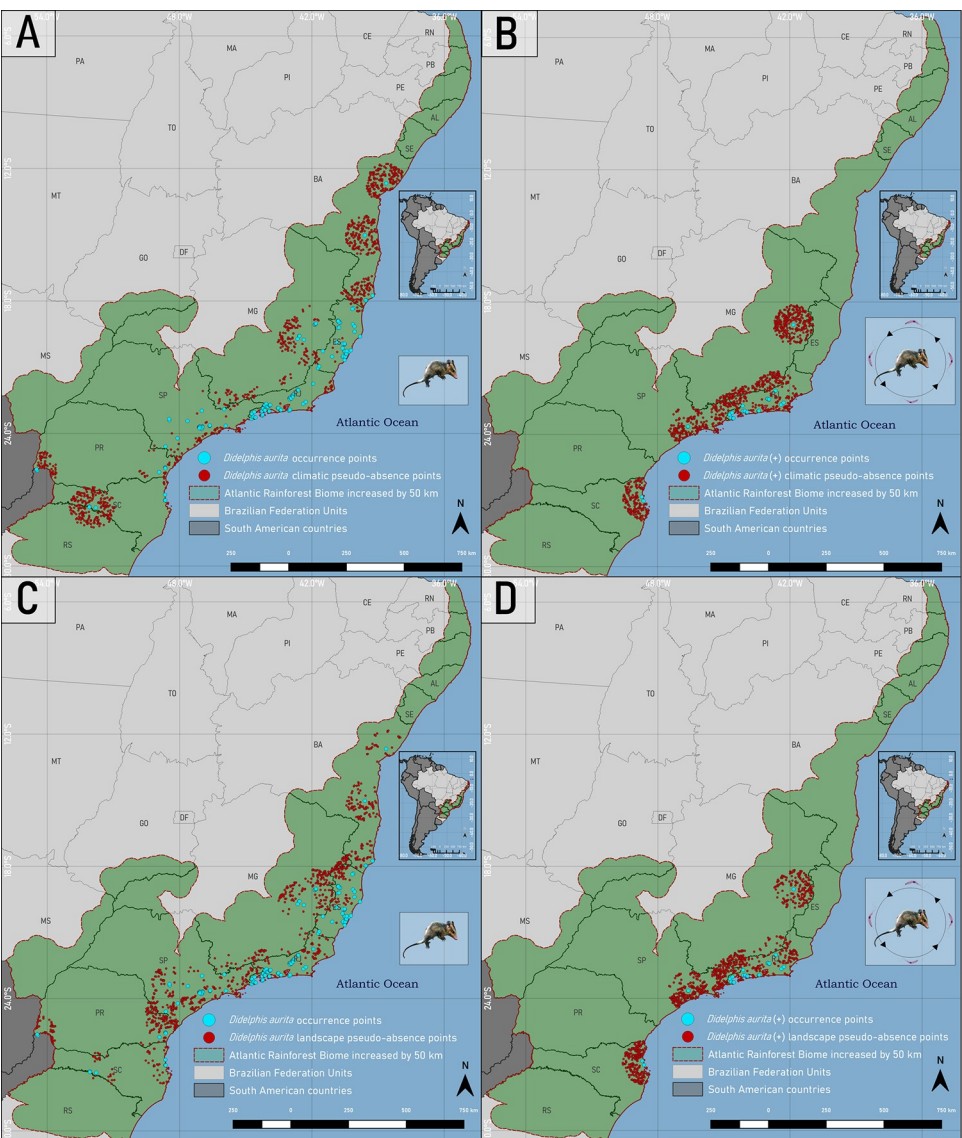

**Fig 3.** Distribution map of the occurrence and pseudo-absence points of *Didelphis aurita* (A) and its positive hemoculture (B) in the climatic approach, and of *Didelphis aurita* (C) and its positive hemoculture (D) in the landscape approach, in the Atlantic Rainforest biome. Software: QGIS 3.22. Source: 1. Instituto Brasileiro de Geografia e Estatística (IBGE). Continuous cartographic base at 1:1,000,000 scale (2016). Available from: https://www.ibge.gov.br/geociencias/cartas-e-mapas/bases-cartograficas-continuas/15759-brasil.html?edicao=16033&t=downloads; Biomes and Coastal-Marine System of Brazil at 1:250,000 scale. Available from: https://www.ibge.gov.br/geociencias/cartas-e-mapas/informacoes-ambientais/15842-biomas.html?=&t=downloads.

0.48–0.72 and 0.52–0.71, respectively (Fig 7A and Table 2). The Mahalanobis distance returned high hit rates (0.74–0.9), with an outlier of TSS = 0.63 (Fig 7A and Table 2). For the *T. cruzi* infection model in the climate approach (Fig 6B and S3 Table), the algorithms selected for the modeling, and which returned the best results, were Random Forests, Maxent, BRT and Domain, generating a model per algorithm and, later, the final Ensemble model from the union of 20 partitions for *D. aurita* infected by *T. cruzi* in the climatic approach (Fig 7B and Table 2). The SVM and GLM algorithms presented the cut-off thresholds below 10%, of 2% - 4.5% and 1% - 8.11%, respectively, while the Mahalanobis Distance and Domain returned the

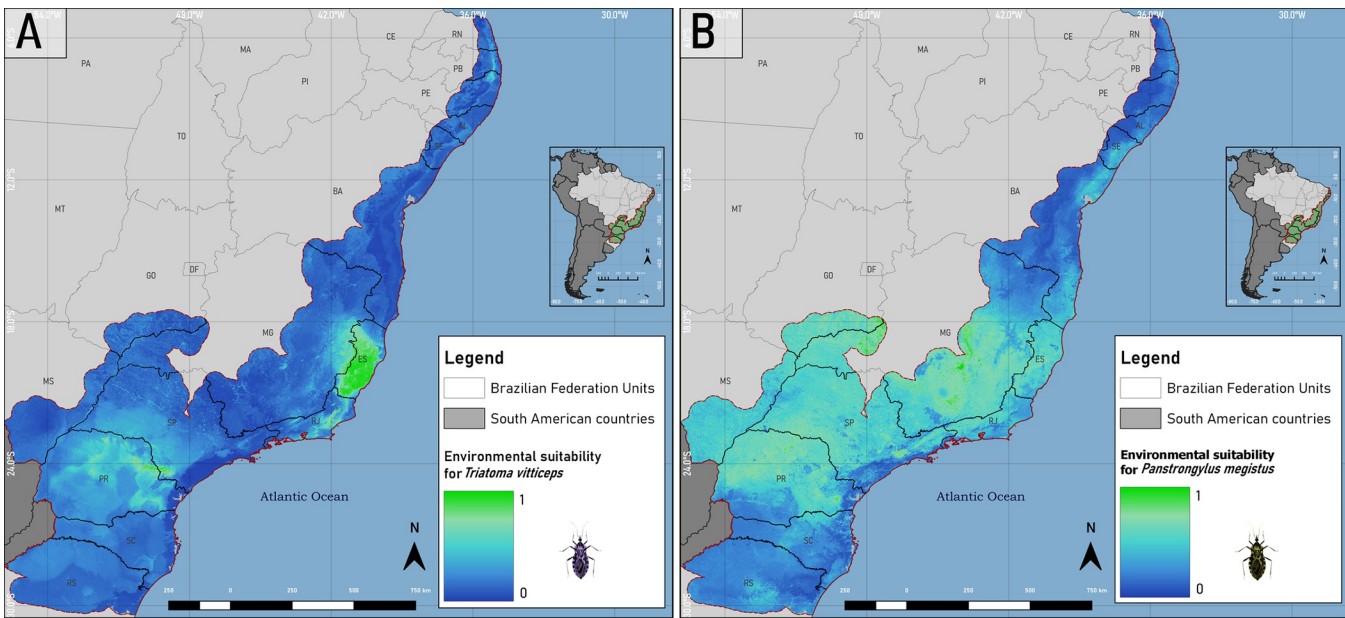

**Fig 4.** Ecological niche models of *Triatoma vitticeps* (A) and *Panstrongylus megistus* (B) in the Atlantic Rainforest biome. Software: QGIS 3.22. Source: 1. Instituto Brasileiro de Geografia e Estatísita (IBGE). Continuous cartographic base at 1:1,000,000 scale (2016). Available from: https://www.ibge.gov.br/geociencias/cartas-e-mapas/bases-cartograficas-continuas/15759-brasil.html?edicao=16033&t=downloads; Biomes and Coastal-Marine System of Brazil at 1:250,000 scale. Available from: https://www.ibge.gov.br/geociencias/cartas-e-mapas/informacoes-ambientais/15842-biomas.html?=&t=downloads.

lowest values of TSS, of 0.167–0.4 and 0.245–0.78, respectively (Fig 7B and Table 2). The SVM, GLM and Mahalanobis Distance algorithms were removed from the modeling, not taking advantage of any of their partitions.

In the landscape approach for *D. aurita*, 63 partitions related to all algorithms were used (Fig 7C), producing the Ensemble model of *Didelphis aurita* in the landscape approach (Fig 6C and S3 Table). The six algorithms SVM, Random Forest, Maxent, GLM, Domain and BRT obtained the best results in the TSS with a variation of 0.80–0.995, with only the Mahalanobis distance algorithm varying from 0.685–0.87, with an outlier of TSS = 0.64, presenting the largest interquartile range between the algorithms, of 0.0674 (Fig 7C and Table 3). For *D. aurita* infected by *T. cruzi* in the landscape approach, the algorithms used for modeling were Random Forest, Maxent, BRT and Domain, generating a model per algorithm and later the Ensemble model (Fig 6D and S3 Table), resulting from the union between 28 partitions (Fig 7D and Table 3). The SVM and GLM algorithms presented the cut-off values below 10%, of 1.3% - 2.4% and 2.6% - 6%, respectively, while the Mahalanobis Distance returned low TSS values, ranging from 0–0.7 (Fig 7D and Table 3).

## Variance partition: Analysis of the contribution of variables

The results obtained through the variance partition indicate that, for *D. aurita* in the climatic approach (S1 Fig), the temperature variables (BIO2, BIO4, BIO5, BIO8 and BIO9) represent the highest percentage of contribution in defining the niche model species, 30% (Adj $R^2$ = 0.3), followed by precipitation (BIO12, BIO13 and BIO15) which is equivalent to 3% (Adj $R^2$ = 0.03). On the other hand, the relationship between the variables of temperature and precipitation presents a degree of contribution in the explanation of the model of 19% (Adj $R^2$ = 0.19). As residuals, that is, percentages that could not be explained by the variables used in the modeling, represent 48% (Adj $R^2$ = 0.48).

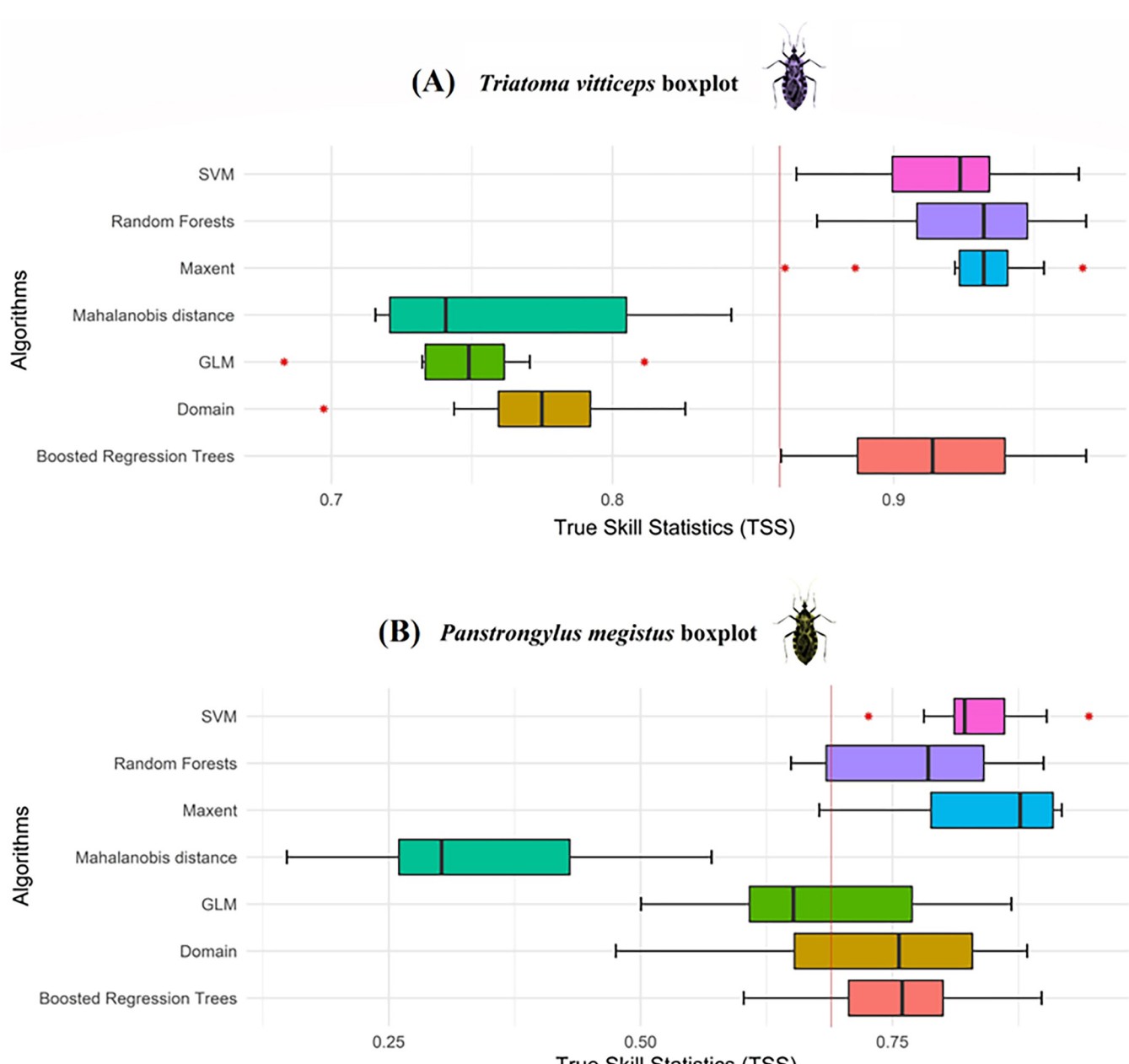

**Fig 5.** Boxplot analysis of True Skill Statistics (TSS) results from *Triatoma vitticeps* (A) and *Panstrongylus megistus* (B) models in the Atlantic Rainforest biome. The red line represents the cut-off value for the TSS of partitions with good statistical quality. Software: RStudio, under R programming language version 4.1.2.

For the ecological niche model of *D. aurita* in the landscape approach, the analysis resulted in a degree of contribution of the Euclidean distance variable to the closest pixel with human presence of 78% (Adj $R^2$ = 0.78), being the highest result of this modeling, followed by the NDVI variable, which explains the model in 21% (Adj $R^2$ = 0.21), while the Euclidean distance to the nearest drainage section is equivalent to 6% (Adj $R^2$ = 0.06) of explanation. The three variables together represent 1% of contribution (Adj $R^2$ = 0.01), and the residuals present in the model are equivalent to 15%.

**Table 1. TSS performance ranges and their interquartile ranges by algorithms for modeling *Triatoma vitticeps* and *Panstrongylus megistus*.**

| Algorithms | *Triatoma vitticeps* | | *Panstrongylus megistus* | |
|---|---|---|---|---|
| | Performance Range | Interquartile Range | Performance Range | Interquartile Range |
| SVM | 0.87–0.97 | 0.0345 | 0.73–0.95 | 0.0497 |
| Random Forest | 0.87–0.97 | 0.0393 | 0.65–0.90 | 0.1563 |
| Maxent | 0.86–0.97 | 0.0171 | 0.68–0.92 | 0.1211 |
| Boosted Regression Trees | 0.86–0.97 | 0.0524 | 0.60–0.90 | 0.0934 |
| Mahalanobis Distance | 0.72–0.84 | 0.0841 | 0.15–0.57 | 0.1695 |
| GLM | 0.68–0.81 | 0.0281 | 0.50–0.87 | 0.1614 |
| Domain | 0.70–0.83 | 0.0327 | 0.48–0.88 | 0.1767 |

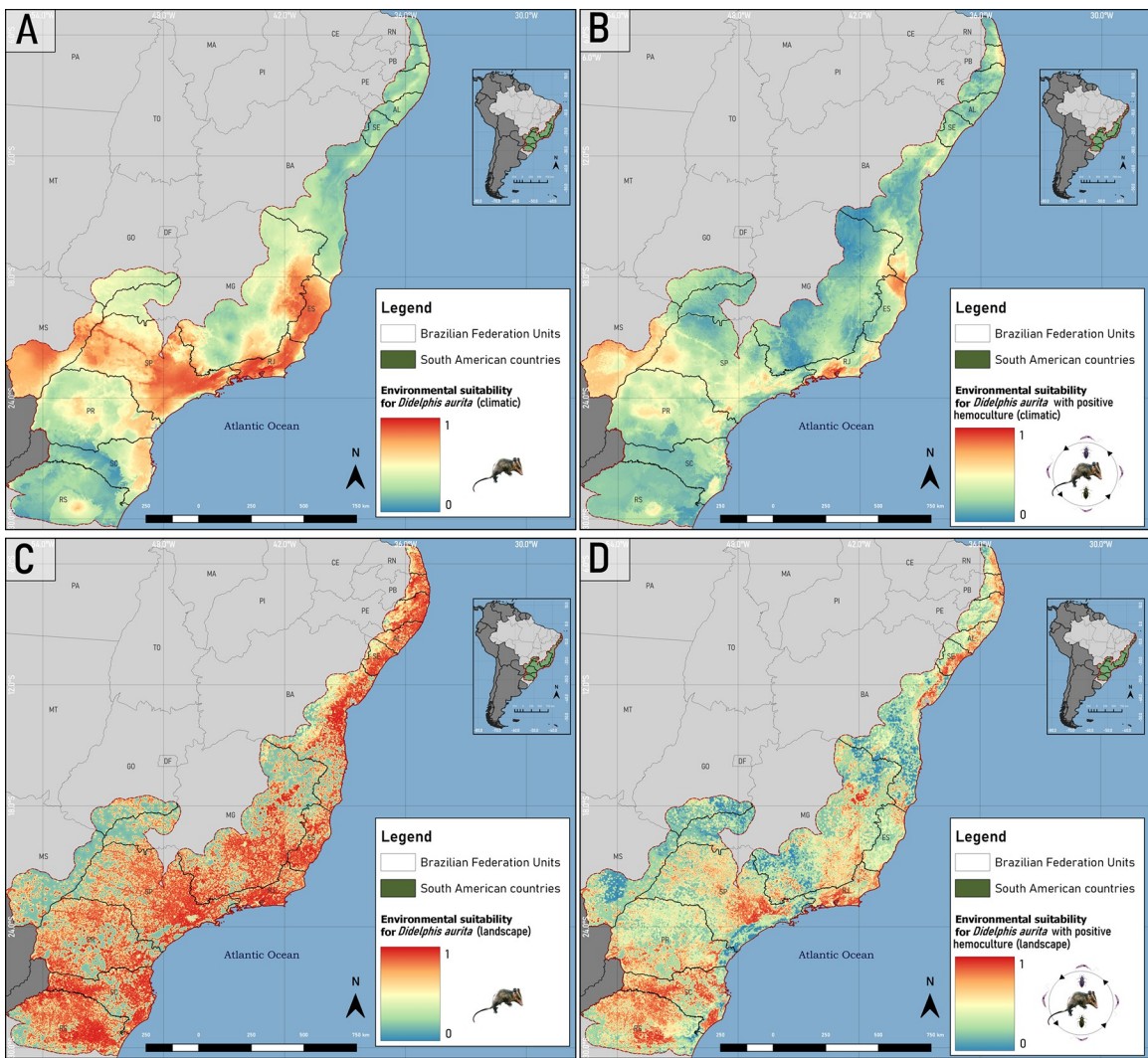

**Fig 6.** Ecological niche models of *Didelphis aurita* (A) and its positive hemoculture (B) in the climate approach, and of *Didelphis aurita* (C) and its positive hemoculture (D) in the landscape approach, in the Atlantic Rainforest biome. Software: QGIS 3.22. Source: 1. Instituto Brasileiro de Geografia e Estatística (IBGE). Continuous cartographic base at 1:1,000,000 scale (2016). Available from: https://www.ibge.gov.br/geociencias/cartas-e-mapas/bases-cartograficas-continuas/15759-brasil.html?edicao=16033&t=downloads; Biomes and Coastal-Marine System of Brazil at 1:250,000 scale. Available from: https://www.ibge.gov.br/geociencias/cartas-e-mapas/informacoes-ambientais/15842-biomas.html?=&t=downloads.

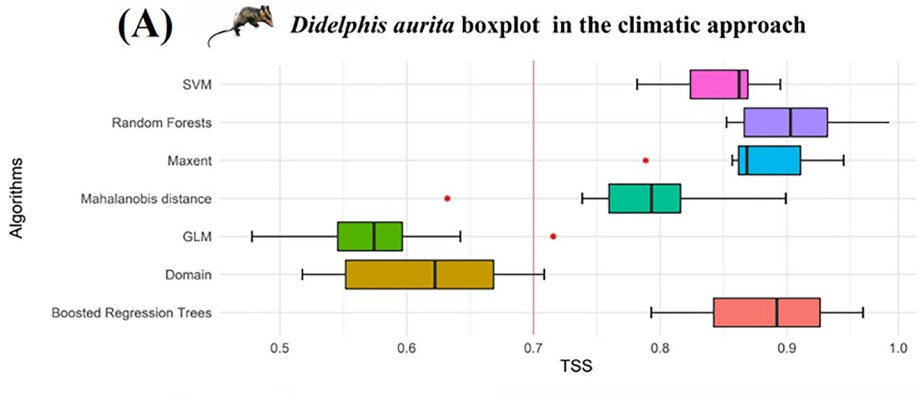

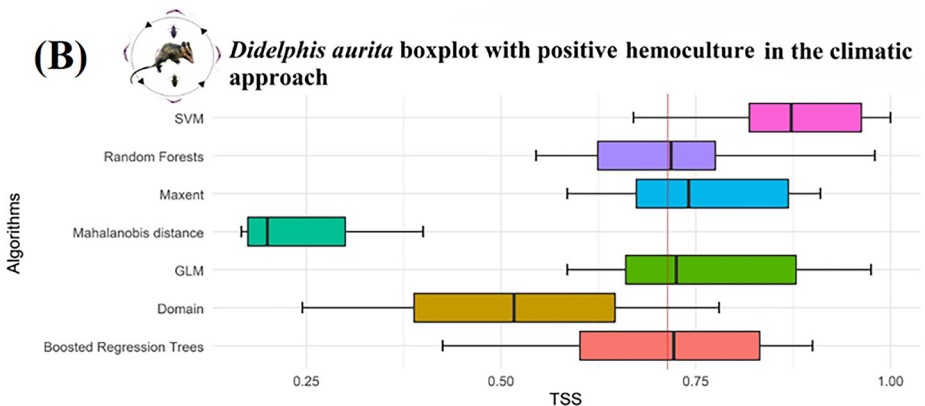

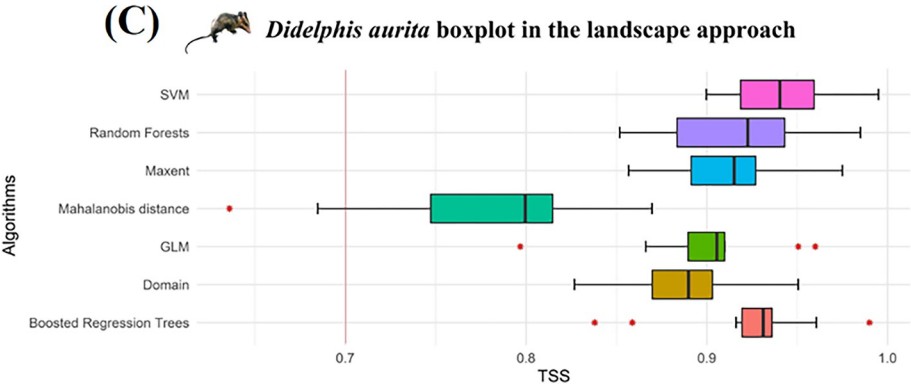

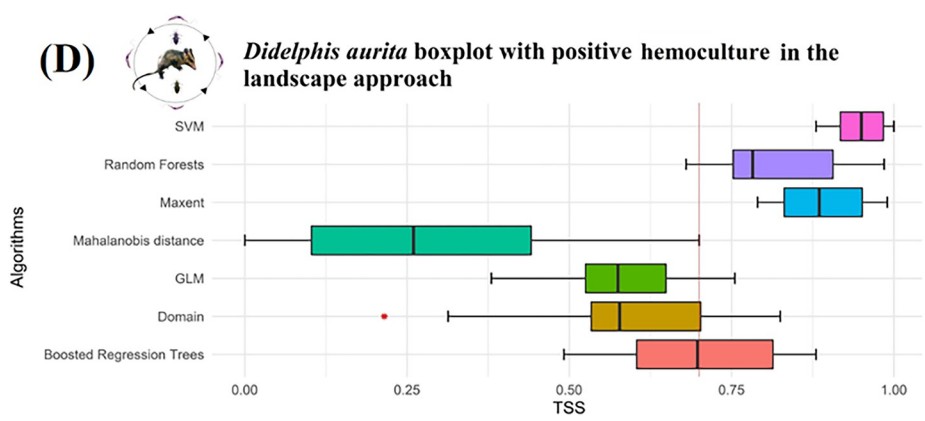

**Fig 7.** Boxplot analysis of True Skill Statistics (TSS) results from ecological niche models of *Didelphis aurita* (A) and its positive hemoculture (B) in the climate approach, and of *Didelphis aurita* (C) and its positive hemoculture (D) in the landscape approach, in the Atlantic Rainforest biome. The red line represents the cut-off value for the TSS of partitions with good statistical quality. Software: RStudio, under R programming language version 4.1.2.

The variance partition for the climatic variables of *D. aurita* infected by *T. cruzi* (positive blood culture) (Fig 8A), in the climatic approach, allowed the analysis of the degree of contribution between the two groups of temperature variables, BIO2, BIO4 and BIO5, and BIO8 and BIO9, and the group of precipitation variables BIO12, BIO13 and BIO15, in relation to the final model generated. In the analysis of the Partition of Variance for *D. aurita* infected by *T. cruzi* (positive blood culture) in the landscape approach (Fig 8B), the analysis of the contribution between the variables EDNPP, EDNDS, NDVI and the triatomine models was performed.

## Ecoland analysis in the Atlantic Rainforest biome

Application of the Ecoland method to climate and landscape models of *D. aurita* (Fig 9 and S4 Table) indicate that the species distribution is centered in the southeastern region of the Atlantic Rainforest biome, in the states of Rio de Janeiro, Espírito Santo and São Paulo, equivalent to an area of 220,878,845 km$^2$ of agreement between high climatic and landscape suitability. The concordance relationship between the areas of high climatic and landscape conditions, associated with the high suitability for the presence of the triatomines *T. vitticeps* and *P. megistus*, are the states of the southeastern region (Espírito Santo, Minas Gerais, Rio de Janeiro, and São Paulo) and south (Paraná), totaling 27,890,688 km$^2$.

In the climatic approach, the regions with more districts with high suitability for *D. aurita* were: Southeast (Espírito Santo, Rio de Janeiro, São Paulo and Minas Gerais), South (Santa Catarina, Paraná and Rio Grande do Sul) and Midwest. (Mato Grosso do Sul). In the landscape approach, all states of the biome had high suitability, and the states of Mato Grosso do Sul and Goiás had the lowest number of districts with high suitability for the landscape. Regarding the high suitability for the presence of triatomines, the regions with the highest numbers of districts in these conditions are the southeast (Espírito Santo, Minas Gerais, Rio de Janeiro and São Paulo) and south (Paraná).

It is possible to verify the variation of environmental suitability, for climate, landscape and presence of triatomines in the districts of each state of the Atlantic Rainforest biome. Regarding the climate approach, the Southeast (Rio de Janeiro, Espírito Santo, São Paulo) and Midwest (Mato Grosso do Sul) regions stand out for their greater climatic suitability, with amplitudes of minimums (Q1–1.5xIQR) and maximums (Q3 + 1.5xIQR) ranging from,

**Table 2. TSS performance ranges and their interquartile ranges by algorithms for modeling *Didelphis aurita* and *Didelphis aurita* with positive hemoculture, in the climatic approach.**

| Climate approach | | | | |
|---|---|---|---|---|
| Algorithms | *Didelphis aurita* | | *Didelphis aurita* positive hemoculture | |
| | Performance Range | Interquartile Range | Performance Range | Interquartile Range |
| SVM | 0.78–0.9 | 0.0452 | 0.67–1 | 0.1437 |
| Random Forest | 0.86–0.99 | 0.0656 | 0.545–0.98 | 0.1508 |
| Maxent | 0.79–0.95 | 0.0488 | 0.585–0.91 | 0.195 |
| Boosted Regression Trees | 0.79–0.96 | 0.0839 | 0.425–0.9 | 0.2308 |
| Mahalanobis Distance | 0.63–0.90 | 0.0564 | 0.167–0.4 | 0.125 |
| GLM | 0.48–0.72 | 0.0508 | 0.585–0.975 | 0.2187 |
| Domain | 0.52–0.71 | 0.1166 | 0.245–0.78 | 0.258 |

**Table 3. TSS performance ranges and their interquartile ranges by algorithms for modeling *Didelphis aurita* and *Didelphis aurita* with positive hemoculture, in the landscape approach.**

| | Landscape approach | | | |
| --- | --- | --- | --- | --- |
| Algorithms | *Didelphis aurita* | | *Didelphis aurita* positive hemoculture | |
| | Performance Range | Interquartile Range | Performance Range | Interquartile Range |
| SVM | 0.9–0.995 | 0.0407 | 0.88–1 | 0.0662 |
| Random Forest | 0.85–0.985 | 0.0596 | 0.68–0.985 | 0.1537 |
| Maxent | 0.86–0.975 | 0.0357 | 0.79–0.99 | 0.12 |
| Boosted Regression Trees | 0.84–0.99 | 0.0165 | 0.492–0.88 | 0.21 |
| Mahalanobis Distance | 0.64–0.87 | 0.0674 | 0–0.7 | 0.3383 |
| GLM | 0.8–0.96 | 0.0202 | 0.38–0.755 | 0.1238 |
| Domain | 0.83–0.95 | 0.0332 | 0.215–0.825 | 0.1688 |

respectively, 0.642–0.904, 0.660–0.838, 0.371–0.876 and 0.590–0.821, with the highest number of outliers in Rio de Janeiro (28 districts) and Espírito Santo (24 districts).

In the landscape approach, the states of the southeast (São Paulo, Rio de Janeiro, Minas Gerais, Espírito Santo), south (Santa Catarina, Rio Grande do Sul and Paraná) and northeast (Sergipe, Rio Grande do Norte, Pernambuco, Paraíba, Bahia and Alagoas) showed high landscape suitability values, with interquartile ranges above 75% suitability, with Minas Gerais having the highest number of outlier districts (317 districts). Finally, regarding the areas of environmental suitability for the presence of triatomines *T. vitticeps* and *P. megistus*, the state with the greatest suitability was Espírito Santo, with amplitudes of minimum (Q1–1.5xIQR) and maximum (Q3 + 1.5 xIQR) ranging from 0.355–0.899, and interquartile range ranging from 0.619–0.806, with only two outlier districts.

The Ecoland results for *D. aurita* with positive blood culture (Fig 10 and S5 Table) indicate that the centrality of the presence of the infection in the marsupial is present in the southeastern region of the Atlantic Rainforest biome, but with areas of agreement for high suitability

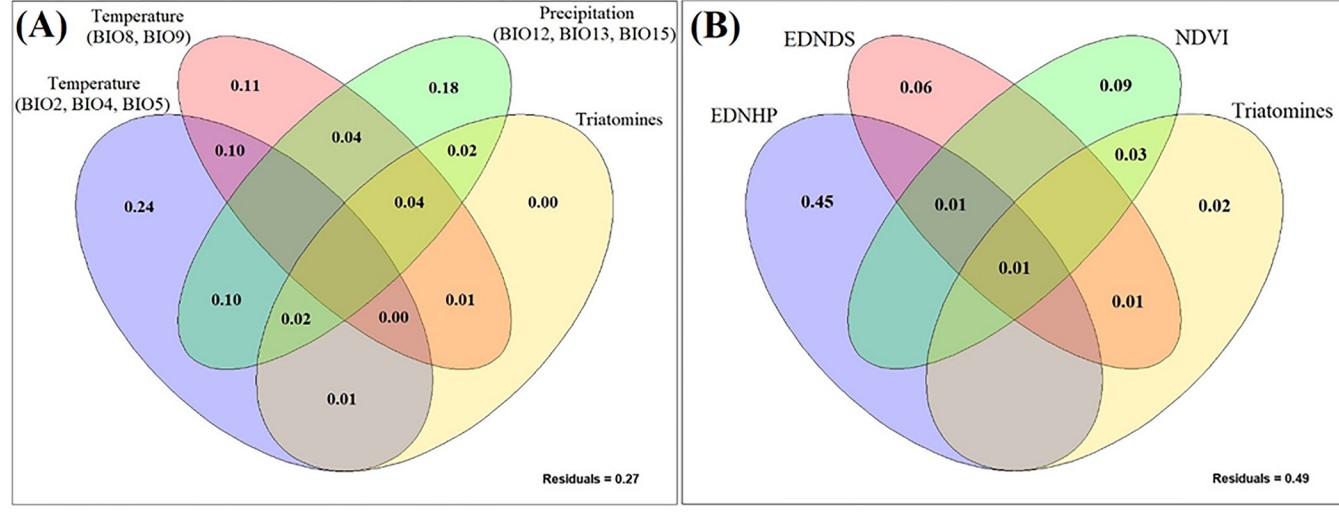

**Fig 8.** Partition of variance: (A) climatic variables of the modeling of *Didelphis aurita* with positive hemoculture for the Atlantic Rainforest biome; (B) landscape variables of the modeling of *Didelphis aurita* with positive hemoculture for the Atlantic Rainforest biome. Standardized data (scale x to zero mean and unit variance), and buffer 0.1˚. Values < 0 note shown, and are interpreted as zeros, corresponding to cases where explanatory variables explain less variation than normal random variables [43]. EDNPP: Euclidean Distance to the Nearest Population Presence; EDNDS: Euclidean Distance to the Nearest Drainage Section. Software: RStudio, under R programming language version 4.1.2.

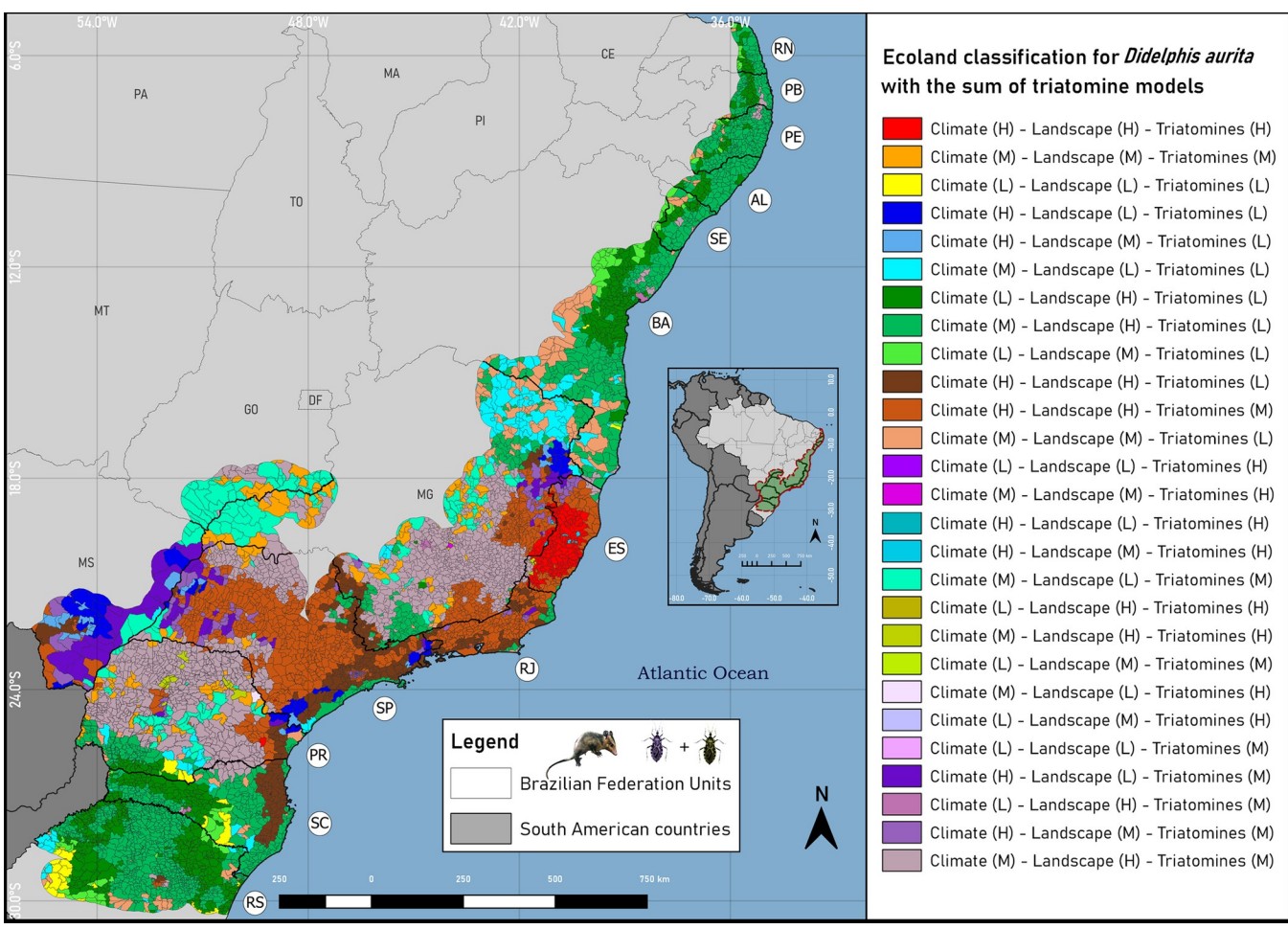

**Fig 9. Ecoland classification map for *Didelphis aurita* in the Atlantic Rainforest biome.** Software: QGIS 3.22. Source: 1. Instituto Brasileiro de Geografia e Estatística (IBGE). Continuous cartographic base at 1:1,000,000 scale (2016). H: High suitability; M: Medium suitability; L: low suitability. Available from: https://www.ibge.gov.br/geociencias/cartas-e-mapas/bases-cartograficas-continuas/15759-brasil.html?edicao=16033&t=downloads; Biomes and Coastal-Marine System of Brazil at 1:250,000 scale. Available from: https://www.ibge.gov.br/geociencias/cartas-e-mapas/informacoes-ambientais/15842-biomas.html?=&t=downloads.

between climate and landscape in a focal way, as in the north and northeast of the state of Espírito Santo (5.39% of the districts), in the Metropolitan region and part of the northeast of Rio de Janeiro (22% of the districts), in Mato Grosso do Sul (3.29% of the districts), Bahia (1.85% of the districts), São Paulo (0.78% of districts), Santa Catarina (0.42% of districts) and Minas Gerais (0.06% of districts) (Fig 10). These suitable areas, climatically and in relation to the landscape, for the presence of positive blood culture, account for 23,832,435 km$^2$ of the Atlantic Rainforest biome.

The areas of medium climate and landscape suitability in agreement with each other, unlike the Ecoland for *D. aurita*, which had 320,700,903 km$^2$ for climate and landscape classified as high suitability and 129,888,271 km$^2$ for medium suitability in climate and landscape. For the Ecoland of *D. aurita* with positive blood culture, an opposite profile was observed, so that the areas of high suitability between both approaches became 23,832,435 km$^2$, and of medium suitability of 250,792,411 km$^2$. The state with the largest number of districts with medium suitability in climate and landscape is Paraná, with 53.07% of its districts in these conditions, occupying the largest area of the biome, of 98,749,298 km$^2$. The districts of the Atlantic Rainforest

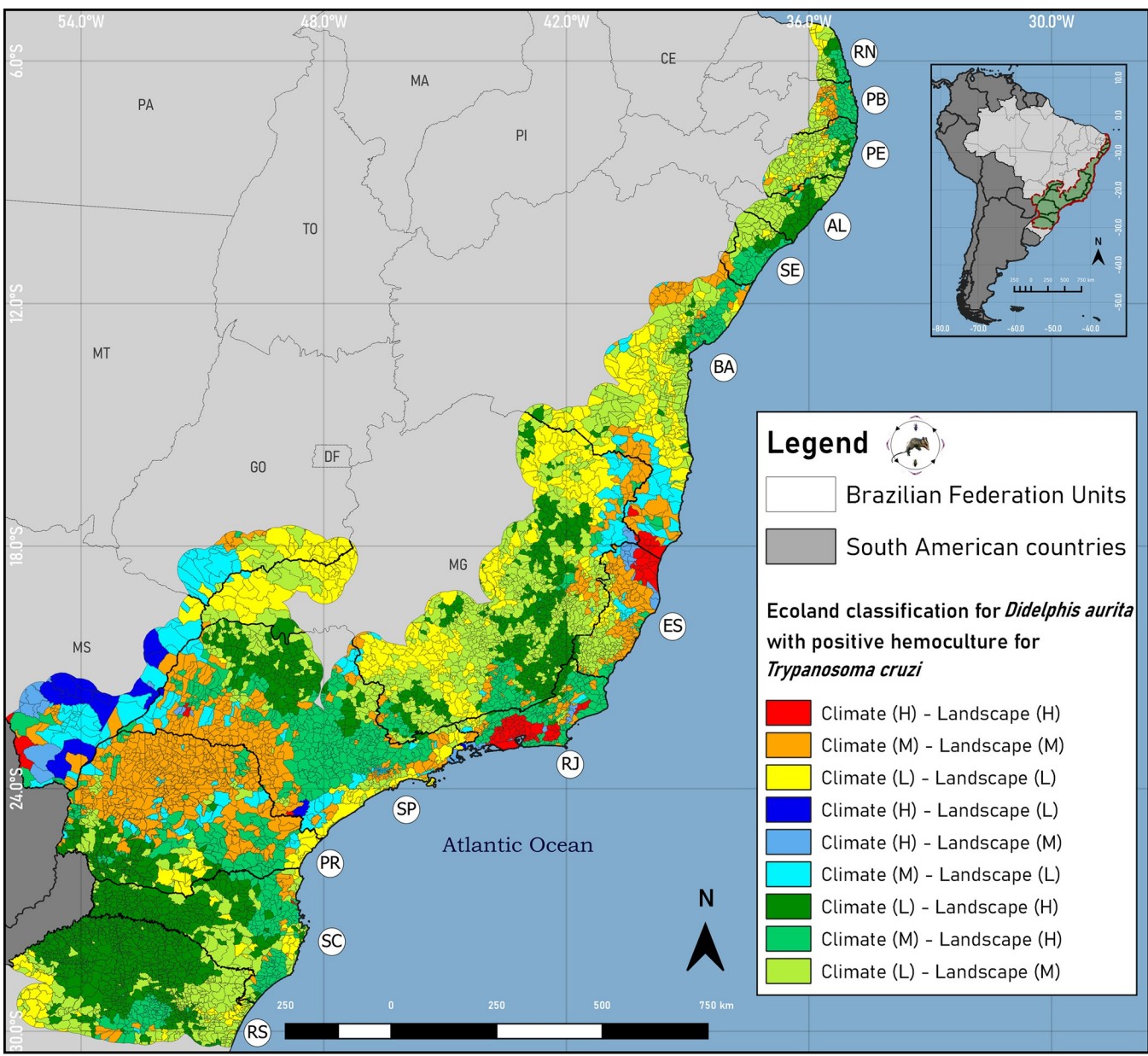

**Fig 10. Ecoland classification map for *Didelphis aurita* with positive hemoculture in the Atlantic Rainforest biome.** Software: QGIS 3.22. Source: 1. Instituto Brasileiro de Geografia e Estatística (IBGE). Continuous cartographic base at 1:1,000,000 scale (2016). H: High suitability; M: Medium suitability; L: low suitability. Available from: https://www.ibge.gov.br/geociencias/cartas-e-mapas/bases-cartograficas-continuas/15759-brasil.html?edicao=16033&t= downloads; Biomes and Coastal-Marine System of Brazil at 1:250,000 scale. Available from: https://www.ibge.gov.br/geociencias/cartas-e-mapas/informacoes-ambientais/15842-biomas.html?=&t=downloads.

biome that are characterized as having low climate and landscape suitability for *D. aurita* total an area of 15,884.58 km$^2$, while for *D. aurita* with positive blood culture this area grew approximately 14.77 times (137%), equivalent to an area of 234,469,636 km$^2$.

Observing the climate approach for *D. aurita* with positive blood culture, the states with the greatest climatic suitability were Mato Grosso do Sul (31.86% of districts), Rio de Janeiro (29% of districts), Espírito Santo (10.07% of districts), Bahia (1.85% of districts), São Paulo (1.57% of

districts), Santa Catarina (0.42% of districts) and Minas Gerais (0.21% of districts), accounting for a total biome area of 72,429,562 km$^2$. In relation to the landscape approach, all states obtained high suitability in this approach, highlighting the three largest: Santa Catarina (77.35% of districts), Rio de Janeiro (73.66% of districts) and Rio Grande do Sul (71.53% of districts).

The marsupial with positive blood culture in the climate approach, the state that stands out the most is Rio de Janeiro, with minimum (Q1–1.5xIQR) and maximum (Q3 + 1.5xIQR) amplitudes of 0.176–0.961, with a difference of 0.1175 in relation to the second highest climatic suitability of the biome, belonging to the district of Itaúnas, in the municipality of Conceição da Barra, in the state of Espírito Santo, which has an suitability of 84.35%. Rio Grande do Sul, with 233 outliers, was the state with the highest number of outlier districts, minimum (Q1–1.5xIQR) and maximum (Q3 + 1.5xIQR) amplitude of 0.188–0.294, and with upper outliers varying of 0.294–0.577. In the landscape approach, of the 15 states in the biome area, only Mato Grosso do Sul, Goiás, Espírito Santo and Bahia presented interquartile intervals below 0.575 (3rd quartile of Bahia). Specifically in relation to Espírito Santo, it presented the lowest interquartile range (0.1041) of the biome, with upper outliers with landscape suitability between 0.746–0.924.

## Discussion

*Didelphis aurita* is endemic to the Atlantic Rainforest biome and is a generalist species, well adapted to places with high levels of anthropic action. It is considered one of the ancestral host species that can maintain all *T. cruzi* genotypes, a parasite characterized by its extreme genetic heterogeneity. Furthermore, autochthonous cases of CD have now been detected in the Atlantic Rainforest [44,45], probably as a consequence of man increasingly exploring the wild environment. This fact is associated with the decrease in the mammal fauna, which results in the loss of food source for triatomines and the consequent invasion of homes by triatomines attracted by light.

The occurrence points of *T. vitticeps* (Fig 2A) indicates that there is a spatial dependence with warm areas in the southeastern region of the Atlantic Rainforest, mainly in Espírito Santo. The *T. vitticeps* ecological niche ensemble model (Fig 4A) resulted in areas of environmental suitability focused on the southeast of the Atlantic Rainforest biome, which include the states of Paraná, São Paulo, Rio de Janeiro and Espírito Santo (in this state it is distributed in 73% of its territory with suitability (β) $\geq$ 50%). In the available databases there are no collection points in the states of Paraná and São Paulo, indicating that the algorithms predicted the environmental characteristics of the presence points to regions without registration. This means that these areas present suitability for the presence of the species, but that specimens were not collected in these regions (sample void), or that due to some other factor, which is not being addressed in this work, the specimens were not able to reach these locations (geographical barriers). The importance of this result means that, in addition to the environmental variables not being highly correlated ($> 0.7$), the occurrence points are not biasing the model due to their concentration in Espírito Santo. These points are distributed throughout the state, and not in places close to each other, which could lead to redundant climatic and landscape features. Thus, the algorithms predicted the distribution to other states, with potential areas for the presence of the species, which can support the direction of new field collections, helping to identify areas with sampling void and public management for the study of the *T. cruzi* enzootic cycle.

In relation to *P. megistus* (Fig 2B), after the application of spatial filters, despite the amount having been reduced to 8.49% of the total composition of the database, the points are

distributed in a great environmental diversity. By analyzing the generated pseudo-absences, their sampling areas (β ≤ 1.9%) occurred in large regions without sampling of pseudo-absences around the presence points, indicating that they are places with suitability for the presence of the species. That is, just by analyzing this pattern, the Atlantic Rainforest biome presented environmental conditions, in large extensions, for the presence of the species.

The Ensemble model for *P. megistus* (Fig 4B) presented a distribution with high environmental suitability concentrated in the Southeast region of the Atlantic Rainforest biome, encompassing large areas of environmental suitability. Environmental conditions are factors that influence the life dynamics of triatomines, such as relative humidity, which in low percentages stimulates an increase in the frequency of feeding of the specimens to avoid dehydration, and consequently increases the search for a food source [46]. Some of the variables that may influence the life of triatomines are: relative humidity, temperature, precipitation, vegetation and wind speed [46–51]. In the case of vegetation, the NDVI was used because it allows identifying the health and density of vegetation through the algebraic relationship between the spectral responses of the near infrared and the red band, being important factors for the life dynamics of triatomines [49–51].

Regarding the performance of their algorithms (Table 1), they were more dispersed compared to *T. vitticeps*, with larger interquartile ranges. Unlike *T. vitticeps*, in which it was possible to observe agreement between the algorithms, greater variability was observed between them, which may have occurred due to the greater environmental diversity of the distribution of *P. megistus* in the biome, while *T. vitticeps* presents greater geographic and environmental restriction in the southeast of the Atlantic Forest (distribution focus and concentration in the state of Espírito Santo).

*Didelphis aurita* inhabits areas of humid primary and secondary forests of the Atlantic Rainforest biome, and may share areas with *Didelphis albiventris* [52]. *D. aurita* is opportunistic and tolerant to altered environments, being found in highly fragmented landscapes and even on the outskirts of urban centers [53]. However, it is a species more associated with preserved environments than *D. albiventris*, being more sensitive to fragmentation of the landscape, so that it may be becoming extinct in marginal regions of the Atlantic Rainforest in contact with other biomes such as the Pampa and Cerrado, indicating a retraction of this species towards the center of the biome [54,55]. In the Ensemble niche model for *D. aurita* (Fig 6A), the regions with the greatest climatic suitability for the presence of *D. aurita* were the Southeast and South regions of the biome (S3 Table), while the states of the Northeast of the biome presented areas of low to medium climatic suitability, with a total variation of 22% to 65% (S3 Table), with their highest value equal to 0.52, located in Bahia. These low values of climatic suitability do not mean that the presence of *D. aurita* does not occur in these areas, but that they are areas with less favorable climatic conditions for the presence of the species, compared to the southeast of the biome. The Ensemble model produced for *D. aurita* in the landscape approach (Fig 6C) presents areas of high landscape suitability (β ≥ 65.8) in a large extension of the Atlantic Rainforest biome (56% of the biome), with higher concentrations in the states Southeast, South and Northeast (S3 Table). The distribution of suitability areas was not homogeneous in the geographic space, unlike the climatic approach, which has a gradual variation from one pixel to another.

The temperature variables (BIO2, BIO4, BIO5, BIO8 and BIO 9) were the ones that most impacted the modeling of the *D. aurita* niche, while precipitation alone contributed with only 3% to the modeling (S1 Fig), and this is a scenario that is related to the characteristics of the species, since temperature is one of the factors that influence the beginning of the breeding season, while precipitation, although it does not have a major impact on the model, does influence the availability of food [54].

As for the landscape approach, to model the habitat of *D. aurita* (Fig 6C), the variables NDVI, Euclidean Distance from an area to the nearest places with human presence and the Euclidean Distance from an area to the nearest drainage stretch were used. The use of these variables was important to understand the habitat of *D. aurita* because it is a generalist species with the ability to occupy different forest strata, and inhabiting areas of intense anthropic action [54,56,57]. In landscape modeling, the variable of proximity to areas of human presence was the one that most impacted (78%) the contribution (S1 Fig) to the definition of the habitat of the species, in which the points of occurrence were present in areas close to human presence. The NDVI occupied the second greater contribution of the modeling, and this may be a consequence of the fact that this index reflects the health and density of the vegetation. The NDVI did not obtain intersection contributions with the proximity variable to human presence, only when they were associated together with the presence of drainage stretches. This indicates that the NDVI and the EDNHP present different interpretation profiles of the model, even though the vegetation index also reflects the urban presence in its spectral composition. As the spatial resolution of this work is 30 arc-seconds (~1km$^2$), the average of the spectral responses of the targets present in each pixel will define its value as a function of the predominant response, in such a way that an area of 1 km$^2$ defined as high values of NDVI, that is, a dense and healthy vegetation, in reality it can also have human constructions inside it, emphasizing the difference in scale and interpretative profile between both variables.

An important point to note is that the landscape modeling was performed with a median of the NDVI in the 2007–2020 range for the entire biome, and this index reflects the health of the vegetation at the time of image collection by the orbital sensor and in the density of vegetation. This indicates that the NDVI variable does not have a fully directed relationship with the identification of vegetation species, but with the spectral reflectance of their chlorophyll [58]. Thus, the relationship of this variable with the points of presence of *D. aurita* reflects in areas with similar vegetation health in which the individuals were located, and which, associated with the proximity to areas of human presence and drainage stretches, returned an area of extensive distribution. In addition, the NDVI, being the median of 13 years of image collection, represents the most frequent pixels in each occurrence location, but does not necessarily represent the characteristic of the location on the date of collection of the specimen point, and this situation is precisely mitigated by using the median and spatial resolution of 1 km$^2$, generalizing the areas.

There are reports of triatomines cohabiting with the genus *Didelphis*, as in the case of *P. megistus*, due to its predilection for hollow trees in arboreal environments inhabited by these marsupials [16], while there are reports of *T. vitticeps* forming colonies associated with opossum nests in peridomiciliary areas in the Atlantic Rainforest [59]. Therefore, it was essential to include predictive models of *T. vitticeps* and *P. megistus* in the modeling of *D. aurita* infected by *T. cruzi*. However, an important point that must be highlighted is that we worked with data from positive blood culture from *D. aurita*, which reflects the presence of infective *T. cruzi* parasites in hosts' bloodstream. This represents a temporal window (detectable parasitemia), directly related to the transmissibility potential of that host, which can vary in time and space. Thus, although the model indicates that a region has high suitability of areas for the presence of opossums infected by *T. cruzi*, this condition is variable in time and space, and the diversity of the local fauna still needs to be considered, in which other hosts can participate in the transmission cycle in nature. As it is a multi-host parasite involved in complex and dynamic cycles of sylvatic transmission, these factors must be considered during fieldworks.

After the application of spatial filters, the points of presence of *D. aurita* infected by *T. cruzi* add up to 27 records (Fig 3B and 3D). Two fundamental points should be mentioned about them: i) environmental diversity due to geographic extension; and ii) consequence of the low quantity in the modeling results. For the first point, the occurrences were distributed in 4

states. In the second point, what would make the modeling impossible would be the low quantity associated with a low environmental diversity. However, in this work, even though the quantitative is only 27 points, the environmental diversity of distribution in four states of the biome allowed obtaining results with hit rates that enable the applicability of the ecological niche model of *D. aurita* infected by *T. cruzi*.

In a comparison in the southern and southeastern regions of the biome between the ecological niche models of *D. aurita* infected by *T. cruzi* (Fig 6B) and the ecological niche model of the species *D. aurita* (Fig 6A), as expected, the areas of climatic suitability for the presence of infection in the marsupial were lower and, in most regions, included in the distribution areas of the species *D. aurita*. In the state of Santa Catarina, the high suitability ($\beta \geq 61.24\%$) of distribution of *D. aurita*, is concentrated in coastal regions, while for positive blood cultures, these patches were reduced to an area of 385 km$^2$ ($\beta \geq 65.2\%$), and medium and high suitability of 25,908 km$^2$ ($\beta \geq 32.60\%$), equivalent to 20.8% of the territory. This indicate that, in this state, *D. aurita* acts in the transmission of *T. cruzi* in a focused way for high climatic suitability, in 0.31% of the state territory.

In Espírito Santo, despite being a state with high suitability for the presence of opossums in 89.41% of its area, high climatic suitability was indicated for the presence of *T. cruzi* infection in *D. aurita* in 10,430 km$^2$ (18.26% of the state). By isolating the areas with high and medium suitability of the North Coast and Northwest mesoregions from the rest of the state, the areas of high suitability ($\beta \geq 65.2\%$) in the state become only 10 km$^2$ (0.02435% of the state), and of medium and high suitability ($\beta \geq 32.60\%$) of 19,790 km$^2$ (48.18% of the state). However, although 48.18% of the state is almost half of this isolated area, the interquartile range of climatic suitability ranged from 0.267 to 0.369, indicating that the predominance of suitability in this region is closer to low suitability, with the presence of 1480 km$^2$ of outliers of minimums and maximums from 0.52 to 0.68.

Dario et. Al. [59] carried out a study to investigate a case of acute Chagas disease that occurred in the municipality of Guarapari in Espírito Santo, through oral transmission from the accidental ingestion of feces from a dead and infected *T. vitticeps*. One of its objectives was to detect the DTUs related to this patient, who was infected by the TcI, TcII, TcIII and TcIV genotypes. Later, Dario et. Al. [60] hypothesized that, in the region, bats act as the main reservoirs of *T. cruzi*, while *T. vitticeps* is the main triatomine vector, described with simple and mixed *T. cruzi* infections. Although *D. aurita* were captured in the area, only one presented positive result in the serological diagnosis and all were negative in blood cultures. These results are in accordance with the results obtained in the ecological niche and habitat models of *D. aurita* with positive blood culture, highlighting that, possibly, *D. aurita* does not have the potential for transmitting *T. cruzi* in the region, and probably other hosts participate in the transmission cycle of *T. cruzi* in that area.

In the northeast region, in the states of Pernambuco, Paraíba and Rio Grande do Norte, few areas (196 km$^2$) demonstrated high suitability for *T. cruzi* infection in *D. aurita*, while there were no areas with high suitability for the presence of the marsupial in these areas ($\beta \geq 61.24\%$). Altogether, there are 17,383 km$^2$ of area with medium and high suitability ($\beta \geq 32.60\%$) for positive blood culture, with 45,529 km$^2$ for the distribution of *D. aurita* in these suitability ($\beta \geq 32.60\%$), with several regions with better climatic condition for the presence of the infection than for the presence of the marsupial. That is, for the opossums that exist in these areas, there are favorable climatic conditions to have a positive blood culture. The same phenomenon was observed in Espírito Santo and Bahia (South Bahian). We may not exclude that these situations indicate model commission errors. However, of the 4 resulting algorithms that were used to generate the Ensemble model, three (Maxent, Random Forest and BRT)

predicted these areas with high suitability for the infected marsupial, while only Domain indicated low suitability. This means that the possibility of it being a commission error is low.

The areas that present the greatest climatic suitability for *D. aurita* with positive blood culture, according to the IBGE climate map of Brazil, are: i) Hot areas (average > 18˚C in all months) humid, sub-dry and super-humid; ii) Sub-Hot areas (average between 15 and 18˚C in at least 1 month) humid; and iii) transition areas between the Subwarm (average between 15 and 18˚C in at least 1 month) super-humid (average between 10 and 15˚C) super-humid.

Of the variables used to model the ecological niche of *D. aurita* with positive blood culture, the temperature variables prevailed in the explanation of the model (Fig 8A). Superficially–and simply–this information is leading to the idea that the presence of positive blood culture in marsupials depends on the life dynamics of triatomines as a function of climatic relationships in space. However, the parasite is deeply involved in a trophic network, and the opossum does not necessarily depend on vector-contamination routes and/or predation by triatomines to become infected and may acquire the infection through the predation of infected small mammals.

Considering the analysis of the climatic variables of the model of *D. aurita* and *D. aurita* with positive blood culture, the residues dropped from 48% to 27%, while for the landscape the residues increased from 15% to 49% (Figs 8 and S1). For *D. aurita* infected by *T. cruzi* (Fig 8B), the variable of distance to the nearest human presence continued to be the largest contribution of the model (45%), followed by NDVI (9%), proximity to rivers (6%) and triatomines (2%). This indicates that areas close to human dwellings that have dense and healthy vegetation are the regions that favor the presence of infected opossums. As a hypothesis, even though an analysis of landscape metrics was not carried out in this work, the areas that have fragments of landscape and/or that are border areas to the presence of human habitation are the ones with the highest chance of finding *D. aurita* with positive blood culture. This is already a specific characteristic of the opossum itself, not a finding in relation to it or its infection.

Regarding the variable of proximity to rivers, although the NDVI and proximity to human presence have reduced their contributions in the model, it did not remain at 6%, as well as its intersection with the two variables mentioned (1%). This indicates that sites close to rivers are areas of importance for the presence of the marsupial, regardless of the presence of infection. This is a factor that can be strategic for the survival of the individual, mainly due to the characteristic of vegetation areas with high NDVI values. Returning to the aforementioned hypothesis, as these rivers are border areas and even in certain types of vegetation fragments, and depending on the city and location, these rivers may not be polluted, being of strategic interest for the animal's subsistence. However, even being polluted, it is still an important location due to the presence of small mammals as a food source. In relation to triatomines, the total contribution, including their relationship with the other variables, is equivalent to 7%, mainly in relation to the NDVI (3%). This higher value in relation to NDVI may be because triatomines were also modeled using this same variable, and vegetation can influence the life dynamics of triatomines.

In the *D. aurita* habitat model with positive blood culture (Fig 6D), a reduction in the area of landscape suitability in relation to the marsupial habitat model was observed in all states. The landscape model showed more extensive responses to environmental conditions in the biome than the climate models, being mainly fragmented in geographic space, in an irregular way.

For the states of Rio de Janeiro, Minas Gerais, Espírito Santo and Bahia, the percentages of areas with high suitability of landscape of the blood culture model (β ≥ 66.21%) in relation to the landscape model of *D. aurita* (β ≥ 65.8%) were 28.61%. In RJ the reduction of areas with high conditions of infection landscape in relation to those of *D. aurita* was not sudden (39% reduction), while in MG the reduction was 29.81%, in Espírito Santo 86.31% and in Bahia 82.2%. This indicated that, despite the high suitability of the landscape for *D. aurita*, the areas

that are likely to harbor infected hosts are much more specific in landscape fragments, mainly in Espírito Santo and Bahia, and excepting Rio de Janeiro. Espírito Santo and Bahia presented the largest reductions in areas, a phenomenon that agrees with the models of the climate approach. In Espírito Santo, despite being highly suitable for the presence of *D. aurita*, it does not present favorable climatic and landscape conditions for the marsupial to be found with positive blood culture.

For the states of the northeast region of the Atlantic Rainforest biome (Bahia, Sergipe, Alagoas, Pernambuco, Paraíba and Rio Grande do Norte), unlike the climatic approach, all coastal states have high landscape suitability for the presence of *D. aurita*, totaling 178,246 km$^2$ (β ≥ 66.21%). However, the areas with positive blood culture were reduced, but still maintaining large patches with suitability in the Northeast, totaling 43,866 km$^2$, 24.61% in relation to the *D. aurita* model. Thus, the landscape model obtained contradictory results in relation to the climate of the northeast of the biome.

As expected, the landscape models returned non-concordant results in a large part of the Atlantic Rainforest in relation to the climate, since its variables are not limited in space as are those the climate, which changes according to latitude, for example. Thus, the climate model and the landscape complement each other, as the climate provides border areas for the distribution of *D. aurita* and its infection, with homogeneously changes in their values and in smooth climatic gradients from pixel to pixel, while the landscape allows viewing areas with vegetation conditions, human proximity and rivers suitable for the presence of the species and infection, in a fragmented and heterogeneous way in space.

When comparing both climate and landscape models by the Ecoland method (Figs 9 and 10), there was a simplification of the appropriate environmental characteristics for the marsupial in the biome, changing the analysis scale from the pixel level (30 arc-seconds) to a scale at district level. In the municipality of Navegantes in Santa Catarina, for example, in 2005 an outbreak of acute Chagas disease occurred through the oral route (ingestion of contaminated sugarcane juice) [10]. In Ecoland for *D. aurita* (Fig 9) Navegantes shows agreement between climate and landscape for high suitability in the model of *D. aurita*, with averages of 68.5% and 97%, respectively (S4 Table). However, it is an area with low suitability for the presence of *T. vitticeps* and *P. megistus*, 25.04%. This result means that in the region, for these vector species, environmental conditions are low, so that other species are acting as vectors. In fact, *Triatoma tibiamaculata* was the species found in palm trees at the location of the ACD outbreak [10], and this is a species with suitability in the municipality [12].

The Ecoland model for *D. aurita* infected by *T. cruzi*, in Navegantes (Fig 10), was classified with high suitability for the presence of the infection, with an average value of 70.4% for climate and 95.3% for landscape (S5 Table). This indicates that it is a district in which the species *D. aurita*, in addition to being present with high suitability, also acts as a reservoir of *T. cruzi* in the area. Roque et. al. [10] reported that in the area of the outbreak, *D. albiventris* and *D. aurita* were infected by *T. cruzi* in both parasitological and serological assays, confirming that these species participated in the transmission cycle, signaling this region with high transmissibility.

## Conclusion

Climate and landscape models and the Ecoland method are tools that can be worked together to improve the ENM results. Observing the advantages of each of these products, i) the Ecoland analyzes allowed classifying the biome districts by their composition, in their limits, of the prevailing climatic and landscape conditions; ii) the climate models presented a spatial definition of the ecological niche of the species, with variations in suitability occurring in a homogeneous gradation from one pixel to another; and iii) and the landscape models showed a more extensive

distribution of habitat conditions along the Atlantic Rainforest, but due to their fragmented profiles and heterogeneous variation of suitability from one pixel to another, they allow the definition of fragments in the geographic space of the niche, species (climate models) and habitat to guide fieldwork sites. Our results showed that areas of high suitability for the presence of marsupials are a necessary, but not sufficient, condition for *D. aurita* to act as a reservoir for *T. cruzi*, in such a way that this scenario can be changed depending on space. and of time.

## Supporting information

**S1 Fig.** Partition of variance: A) climatic variables of the modeling of *Didelphis aurita* for the Atlantic Rainforest biome; B) landscape variables of the modeling of *Didelphis aurita* for the Atlantic Rainforest biome. Standardized data (scale x to zero mean and unit variance), and buffer 0.1˚. Values < 0 note shown, and are interpreted as zeros, corresponding to cases where explanatory variables explain less variation than normal random variables [43]. EDNPP: Euclidean Distance to the Nearest Population Presence; EDNDS: Euclidean Distance to the Nearest Drainage Section. Software: RStudio, under R programming language version 4.1.2.
(TIF)

**S1 Table. Exclusion and inclusion pseudo-absence buffers for each database.**
(XLSX)

**S2 Table. Environmental variables used to model *Triatoma vitticeps* and *Panstrongylus megistus*, and *Didelphis aurita* and its positive hemoculture, in the climate and landscape approach.**
(XLSX)

**S3 Table. Maximum and minimum values per state of the ecological niche model of *Didelphis aurita* in the climatic and landscape approach, and for *Didelphis aurita* infected by *Trypanosoma cruzi* in the climatic and landscape.**
(XLSX)

**S4 Table. Ecoland classification of climatic, landscape and triatomine presence suitability for *Didelphis aurita* by district in the Atlantic Rainforest biome.**
(XLSX)

**S5 Table. Ecoland classification of climatic, landscape and triatomine presence suitability for *Didelphis aurita* with positive hemoculture by district in the Atlantic Rainforest biome.**
(XLSX)

**S1 Appendix. Google Earth Engine code to export the GHS-POP R2015A - GHS population grid, derived from GPW4, multitemporal (1975, 1990, 2000, 2015) to the Atlantic Rainforest area plus 50 km.**
(PDF)

**S2 Appendix. Google Earth Engine code to export the MOD13A2 V6 product, NDVI band, to the Atlantic Rainforest area plus 50 km.**
(PDF)

## Acknowledgments

We would like to thank Bruno Alves Silva, Allison de Araújo Fabri, Vitor Antônio Louzada de Araújo (Laboratório de Biologia de Tripanosomatídeos IOC/Fiocruz), Camila Lucio,

Fernando de Oliveira Santos, and Sócrates Fraga da Costa Neto for the participation on the fieldwork expeditions, Dr. Paulo Sérgio D'Andrea and Bernardo Rodrigues Teixeira (Laboratório de Biologia e Parasitologia de Mamíferos Silvestres Reservatórios, IOC/Fiocruz); to Carlos Ardé and Marcos Antônio dos Santos Limas (Laboratório de Biologia de Tripanosomatídeos IOC/Fiocruz) for technical support in the hemocultures; we would like to thank the Núcleo de Entomologia e Malacologia from Espírito Santo state health department for providing the triatomine specimens and location information. We also thank Dr Gustavo Rocha Leite from Universidade Federal do Espírito Santo, who provided the Geobase coordinate database and a special thanks to Dr. Vera Bongertz for the English review.

## Author Contributions

**Conceptualization:** Raphael Testai, Ana Maria Jansen, Samanta Cristina das Chagas Xavier.

**Data curation:** Raphael Testai.

**Formal analysis:** Raphael Testai, Marinez Ferreira de Siqueira, Diogo Souza Bezerra Rocha, Andre Luiz Rodrigues Roque, Ana Maria Jansen, Samanta Cristina das Chagas Xavier.

**Methodology:** Raphael Testai, Marinez Ferreira de Siqueira, Diogo Souza Bezerra Rocha, Samanta Cristina das Chagas Xavier.

**Validation:** Raphael Testai.

**Writing – original draft:** Raphael Testai, Andre Luiz Rodrigues Roque, Ana Maria Jansen, Samanta Cristina das Chagas Xavier.

**Writing – review & editing:** Raphael Testai, Andre Luiz Rodrigues Roque, Ana Maria Jansen, Samanta Cristina das Chagas Xavier.

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
