## [Decision Letter · Decision Letter 0]

5 Apr 2023

PONE-D-23-05941Space-environment relationship in the identification of potential areas of expansion of *Trypanosoma cruzi* infection in *Didelphis aurita* in the Atlantic RainforestPLOS ONE

Dear Dr. Xavier,

Thank you for submitting your manuscript to PLOS ONE. After careful consideration, we feel that it has merit but does not fully meet PLOS ONE’s publication criteria as it currently stands. Therefore, we invite you to submit a revised version of the manuscript that addresses the points raised during the review process.

We look forward to receiving your revised manuscript.

Kind regards,

Everton Falcão de Oliveira, Ph.D

Academic Editor

PLOS ONE

Journal Requirements:

2.  Please change "female” or "male" to "woman” or "man" as appropriate, when used as a noun (see for instance https://apastyle.apa.org/style-grammar-guidelines/bias-free-language/gender)

3. We note that Figure (2, 3, 4, 6, 9, 10) in your submission contain copyrighted images. All PLOS content is published under the Creative Commons Attribution License (CC BY 4.0), which means that the manuscript, images, and Supporting Information files will be freely available online, and any third party is permitted to access, download, copy, distribute, and use these materials in any way, even commercially, with proper attribution. For more information, see our copyright guidelines: http://journals.plos.org/plosone/s/licenses-and-copyright.

1. You may seek permission from the original copyright holder of Figure (2, 3, 4, 6, 9, 10) to publish the content specifically under the CC BY 4.0 license. 

"We would like to thank Bruno Alves Silva, Allison de Araújo Fabri, Vitor Antônio Louzada de Araújo (Laboratório de Biologia de Tripanosomatídeos IOC/Fiocruz), Camila Lucio, Fernando de Oliveira Santos, and Sócrates Fraga da Costa Neto  for the participation on the fieldwork expeditions, Dr. Paulo Sérgio D’Andrea and Bernardo Rodrigues Teixeira (Laboratório de Biologia e Parasitologia de Mamíferos Silvestres Reservatórios, IOC/Fiocruz); to Carlos Ardé and Marcos Antônio dos Santos Limas (Laboratório de Biologia de Tripanosomatídeos IOC/Fiocruz) for technical support in the hemocultures; we would like to thank the Núcleo de Entomologia e Malacologia from Espírito Santo state health department for providing the triatomine specimens and location information. We also thank Dr Gustavo Rocha Leite from Universidade Federal do Espírito Santo, who provided the Geobase coordinate database and a special thanks to Dr. Vera Bongertz for the English review."

"Funding: this study was funded by Fundação Oswaldo Cruz (Fiocruz), Conselho Nacional de Desenvolvimento Científico e Tecnológico (CNPq), Fundação Carlos Chagas Filho de Amparo à Pesquisa do Estado do Rio de Janeiro (FAPERJ), Coordenação de Aperfeiçoamento de Pessoal de Nível Superior (CAPES). AR is financially supported by CNPq/Universal (425293/2018-1) and Jovem Cientistas do Nosso Estado/Faperj (E-26/202.794/2019). AJ is financially supported by CNPq (Bolsista de Produtividade, nível 1A). SX has received financial support from CNPq (MCTIC/CNPq No. 28/2018 - Universal, process number 422489/2018-2), and JCNE/FAPERJ (E-26/201.314/2021), AR and SX Faperj (Apoio a Grupos Emergentes de Pesquisa no estado do Rio de Janeiro, process number E-26/010.002276/2019)."

Reviewers' comments:

Reviewer's Responses to Questions

**Comments to the Author**

1. Is the manuscript technically sound, and do the data support the conclusions?

Reviewer #1: Partly

Reviewer #2: Yes

Reviewer #3: Yes

2. Has the statistical analysis been performed appropriately and rigorously? 

Reviewer #1: Yes

Reviewer #2: Yes

Reviewer #3: Yes

3. Have the authors made all data underlying the findings in their manuscript fully available?

Reviewer #1: Yes

Reviewer #2: No

Reviewer #3: Yes

4. Is the manuscript presented in an intelligible fashion and written in standard English?

Reviewer #1: Yes

Reviewer #2: No

Reviewer #3: Yes

5. Review Comments to the Author

Reviewer #1: Thank you for submitting your study to "Plos ONE".

- page 5, line 196-198: I suggest using to obtain the best components among the variables. Please see the references: 1. (Triatomine and Trypanosoma cruzi discrete typing units distribution in a semi-arid area of northeastern Brazil: https://doi.org/10.1016/j.actatropica.2021.105950) and 2. (Co-occurrence or dependence? Using spatial analyses to explore the interaction between palms and Rhodnius triatomines - https://doi.org/10.1186/s13071-020-04088-0)

- page 5, line 213-215: I suggest using the variables from Ambdata (http://www.dpi.inpe.br/Ambdata/English/index.php) and MapBiomas (https://mapbiomas.org/).

- page 5, line 201: Put the script used in Google Earth Engine.

- page 6, line 216-220: Which MODIS product was used? Was a general average calculated for the period studied, or was it calculated by year or by month? If you have used it per year or per month, I suggest using the PCA.

- page 6, line 239: Describe modeling M1, M2, M3, M4 and M5 or put a reference (table S2).

- page 18, line 423-442: I suggest using Principal Components Analysis (PCA) to obtain the best components among the variables.

- page 23, line 573: What are these restrictions?

- page 23, line 584-585: Example to use data from mapbiomas.

- page 24, line 608-609: Example to use the NDVI PCAs.

Figure and Table

- Put the caption of Figures 4 and 6 on the same scale (0 - 1).

- I recommend placing the caption inside figure 9, as it was done in figure 10. This improves the quality of the images/figures.

- Tables S3, S4, S5 and S6 could be grouped into a single table identifying the relative columns for each variable/group of variables.

- Put the classes in Tables S7 and S8 in English.

I also missed the elevation variable (derived from the SRTM), as the altitude effect in the Atlantic Rainforest can represent an obstacle to the existence of opossums/vectors/parasites.

My suggestion is that this treatment (PCA) be done before making the models.

Reviewer #2: This is a modelling study about the ecological niches of multiple species involved in the transmission of T. cruzi in the Brazilian Atlantic forest biome. The methods are carefully thought and well described, with some specific exceptions that I detail below. The results are original and relevant, but their presentation and discussion can be significantly improved by a major revision of the text. As a reader, I found the text hard to read, which hinders the communication of the main results.

My main concern about the methods is the Ecoland approach. I do not think that its results add much to the whole interpretation of the study, at least not in the current version of the manuscript. The maps on figures 9 and 10 have too many different categories and colours, making it hard to summarise any clear message out of them. The main objective of the study (as stated in the abstract) is achieved by the other results and is properly covered in most of the Discussion, whilst the Ecoland approach is only mentioned in the very last sentences of the section. I suggest rewriting a stronger justification to maintain its results. Aggregating the model outputs by municipality can be extremely relevant to decision-makers, but given the high number of municipalities, it is hard to summarise it in a single map. I suggest keeping them only as supplementary files, as they already are in S7 and S8 tables.

Specific suggestions (line numbers from the pdf version):

14-15: This sentence implies that the authors modelled the niche of the parasite, which does not seem to be the objective of the study.

72: Jansen et al. (missing ref)

91: Be careful with the concept of model extrapolation (here and throughout the text). The general ENM approach aims at prediction within the ranges of the variables in the training set. Whether or not strict extrapolation occurs, it depends on the choice of algorithm and the whole modelling approach. See Qiao et al. 2018 (https://doi.org/10.1111/ecog.03986) for clarification.

182: Please describe the criteria for establishing the radii for each species.

199: Please describe the method to assess the Pearson's correlation? (by using the full set or a sub-selection of pixel values)

225-233: Consider rewriting this paragraph for a better description of the method. Were the occurrences artificially selected if they were associated with NDVI values above 0.2?

239: There is no mention of these acronyms elsewhere in the text. Explain what those models are.

239-241: Give more information about the variance partition procedure. Was this done over the final ensemble models? How to assess variable contribution in the ensemble models, after the predictions were averaged?

550 and 565: Review missing symbol font

552-553: If model extrapolation was not specifically assessed, this could be the correct prediction from the models.

559: Same as before, maybe change “extrapolated” to “predicted”

598-600: This sentence would be better suited in the Results section.

702-710: Rewrite for clarity.

* Overall Discussion section: I recommend adding references to figures when a specific result is being discussed.

Fig 1: "DB" is mentioned in the legend, but it is not in the figure.

Fig 5: What does the vertical red line represents? Please describe it in the legend.

Fig 8: This figure is hard to interpret, given the lack of details about the partition of variance method. What does the values in the intersection between variables mean? What is the difference between an empty intersection and the 0.00 value?

Figs 9 and 10: Too many categories/colours make it hard to summarise or extract any spatial trends from these figures

Supplementary files: Consider adding titles and legends to all tables inside the files.

S2 Table: Did the climatic models (M2) include NDVI and the environmental distances? This would be contradictory with S1 Fig and some parts of the text.

Reviewer #3: This is an interesting and well written manuscript. This work is well addressed and the subject is clearly relevant to the journal. The data collected represent a very impressive and interesting dataset, the models are compelling and the results are original and clearly. The outcomes could contribute to a better understanding of T. cruzi transmission in disturbed environments, where generalist reservoirs such as D. aurita are abundant (see attachment for comments).

6. PLOS authors have the option to publish the peer review history of their article (what does this mean?). If published, this will include your full peer review and any attached files.

Reviewer #1: No

Reviewer #2: No

Reviewer #3: No

---

## [Author Response · Author response to Decision Letter 0]

19 Jun 2023

We are thankful for the opportunity to have our manuscript entitled “Space-environment relationship in the identification of potential areas of expansion of Trypanosoma cruzi infection in Didelphis aurita in the Atlantic Rainforest" (PONE-D-23-05941) evaluated by the editorial board and reviewers of the PLoS ONE. We revised our manuscript, accepted and modified several requested aspects and answered each of the points stated by the three reviewers. Moreover, we attached the two versions of the manuscript: one with highlighted changes (uploaded as a "Revised Article with Changes Highlighted" file) and the clean version of the manuscript (uploaded as the article file). We answered each one of the reviewers separately.

Journal Requirements:

“When submitting your revision, we need you to address these additional requirements.”

“1. Please ensure that your manuscript meets PLOS ONE's style requirements, including those for file naming. The PLOS ONE style templates can be found at https://journals.plos.org/plosone/s/file?id=wjVg/PLOSOne_formatting_sample_main_body.pdf and https://journals.plos.org/plosone/s/file?id=ba62/PLOSOne_formatting_sample_title_authors_affiliations.pdf”

Author's response: The text, figures and tables have undergone revisions and now meeting all the journal's requirements. Furthermore, the abstract was adjusted to 300 words.

“2. Please change "female” or "male" to "woman” or "man" as appropriate, when used as a noun (see for instance https://apastyle.apa.org/style-grammar-guidelines/bias-free -language/gender)”

Author's response: In the text, there are no words "female" or "male".

“3. We note that Figure (2, 3, 4, 6, 9, 10) in your submission contain copyrighted images. All PLOS content is published under the Creative Commons Attribution License (CC BY 4.0), which means that the manuscript, images, and Supporting Information files will be freely available online, and any third party is permitted to access, download, copy, distribute, and use these materials in any way, even commercially, with proper attribution. For more information, see our copyright guidelines: http://journals.plos.org/plosone/s/licenses-and-copyright.”

Author's response: All maps, present in Figures 2, 3, 4, 6, 9 and 10 were adjusted, removing the background that used copyrighted images.

Comments made by Reviewers:

Reviewer #1: We are grateful for all the suggestions made by the Reviewer, especially for the methodological recommendations. Also, thanks for the recommended text fixes, which have been corrected. Your reviews were of great importance for the improvement of this work and our future projects. All line and page references are to the marked version of the text "Revised Manuscript with Track Changes". Below are the specific comments:

“- page 5, line 196-198: I suggest using to obtain the best components among the variables. Please see the references: 1. (Triatomine and Trypanosoma cruzi discrete typing units distribution in a semi-arid area of northeastern Brazil: https://doi.org/10.1016/j.actatropica.2021.105950) and 2. (Co-occurrence or dependence? Using spatial analyses to explore the interaction between palms and Rhodnius triatomines - https://doi.org/10.1186/s13071-020-04088-0)”

Author's response: We are very grateful for the suggestion of the PCA method, which is of great value for works involving Ecological Niche Modeling, but correlation analysis is an important tool for detecting highly multicollinear variables and eliminating redundant ones. The Principal Component Analysis (PCA) method performs the calculation of eigenvectors and eigenvalues to define the maximum variance of the data represented by an eigenvalue. Through a matrix of “X” variables, a covariance matrix “A” is calculated, in which its product with the eigenvector (principal component, with greater variance), is equal to the product of this same eigenvector with its eigenvalue (the greater its value, the greater the variance). That is, the PCA method allows the creation of new variables through the set of variables used through a linear combination between the matrix of variables “X” and their respective weights “w”. By the Lagrange method, the ideal weights “w” is equivalent to the eigenvectors of the covariance matrix of the “X” variables, corresponding to the largest eigenvalue. This will allow defining the first component, which is the eigenvector with the largest eigenvalue, and the second component already represents the vector that is perpendicular to the principal component vector and which also has the second largest variance. That is, this method allows indicating which are the better linear combinations between environmental variables to identify their better relationships with a possible response variable. That is, new variables are created (linear combinations), so that redundant variables can be removed through these eigenvectors. However, the problem with the PCA method is that it has a complex interpretation, while correlation index, in the sense of selecting variables for ecological niche modeling, masterfully and didactically solves the problem of redundancy between environmental variables. The correlation index is one of several ways of selecting variables used by researchers to perform ecological niche modeling, and allows comparative studies [1–4]. In addition to these references, the work https://doi.org/10.1186/s13071-020-04088-0 cited by the Reviewer also used correlation index to perform the selection of environmental variables, filtering for nine non-redundant variables. However, correlation was chosen as the variable selection methodology. In our study, the methodology employed for variable selection was correlation.

1. Ferro e Silva AM, Sobral-Souza T, Vancine MH, Muylaert RL, de Abreu AP, Pelloso SM, et al. Spatial prediction of risk areas for vector transmission of Trypanosoma cruzi in the State of Paraná, southern Brazil. Caccone A, organizador. PLoS Negl Trop Dis. 2018;12[10]:e0006907. 

2. Cáceres NC, de Moraes Weber M, Melo GL, Meloro C, Sponchiado J, Carvalho R dos S, et al. Which Factors Determine Spatial Segregation in the South American Opossums (Didelphis aurita and D. albiventris)? An Ecological Niche Modelling and Geometric Morphometrics Approach. Raia P, organizador. PLOS ONE. 2016;11[6]:e0157723. 

3. Mollalo A, Sadeghian A, Israel GD, Rashidi P, Sofizadeh A, Glass GE. Machine learning approaches in GIS-based ecological modeling of the sand fly Phlebotomus papatasi, a vector of zoonotic cutaneous leishmaniasis in Golestan province, Iran. Acta Trop. 2018;188:187–94. 

4. Marques R, Krüger RF, Cunha SK, Silveira AS, Alves DMCC, Rodrigues GD, et al. Climate change impacts on Anopheles (K.) cruzii in urban areas of Atlantic Forest of Brazil: Challenges for malaria diseases. Acta Trop. 2021;224:106123. 

“- page 5, line 213-215: I suggest using the variables from Ambdata (http://www.dpi.inpe.br/Ambdata/English/index.php) and MapBiomas (https://mapbiomas.org/).”

Author's response: We appreciate the recommendation of the Ambdata (INPE) and MapBiomas data by the Reviewer, which are institutions and platforms of great value for the selection of possible variables for understanding the niche and habitat of the species. However, in our work we used product variables of remote sensor images acquired through the Google Earth Engine platform, which enables a better spatial definition of the images of our area of interest and throughout the time series of our work. Regarding climate variables, Ambdata uses the variables from the Worldclim website, which were used in the modeling of this work. Furthermore, the climate variables present on the Ambdata website use the Worldclim 1.4 version, while on the Worldclim website they already use the 2.0 version. In the case of Mapbiomas, it is present in Google Earth Engine for download, however, the use of categorical variables together with continuous variables in Ecological Niche Modeling can lead to low quality statistical results for the models. Therefore, this type of variable was not utilized in the modeling process.

“- page 5, line 201: Put the script used in Google Earth Engine.”

Author's response: The necessary changes were made, including the codes used in Google Earth Engine to acquire the NDVI MOD13A2 V6 product images (S2 Appendix) and the Population Count, from GHSL: Global Human Settlement Layers, Population Grid 1975-1990-2000-2015 (P2016) (S1 Appendix). Both codes have been entered as supplementary data as Appendix S1 and S2.

“- page 6, line 216-220: Which MODIS product was used? Was a general average calculated for the period studied, or was it calculated by year or by month? If you have used it per year or per month, I suggest using the PCA.”

Author's response: The MODIS product used was the MOD13A2 V6, and the calculation method used, which was for the entire period through the median of pixels depending on the temporal resolution of the sensor (16 days). The PCA method would not be suitable for this case, as we calculated the NDVI for the entire pixel-by-pixel period in a temporal resolution of 16 days, with no need for its application. These corrections have been incorporated into the text, specifically at lines 234 - 243, page 6.

“- page 6, line 239: Describe modeling M1, M2, M3, M4 and M5 or put a reference (table S2).”

Author's response: As suggested, references have been included in Table S2.

“- page 18, line 423-442: I suggest using Principal Components Analysis (PCA) to obtain the best components among the variables.”

Author's response: The main difference between the PCA and RDA methods is that the PCA uses eigenvalues to define the creation of new variables from the linear combination of variables with their respective weights, using the explanatory variables to understand the response variables. In the case of RDA, it performs the opposite process, using a matrix of response variables to understand the explanatory variables. That is, while the PCA method allows explaining which variables better explain a possible relationship between the variables for a response variable, the RDA allows explaining the response variables in relation to the explanatory variables. For this reason, the RDA is mostly applied to understand the degree of contribution of environmental variables in relation to the ecological niche model, so that from the model, one seeks to understand its relationship with the environmental variables used. We appreciate the reviewer's suggestion, however, we utilized the Partition of Variance method to assess the contribution of each environmental variable, as well as their intersection in the Venn diagram, in relation to the response variable, which is the Ensemble model. This analysis was performed using the canonical redundancy analysis (RDA), as described in detail on lines 272-289 on page 7.

- page 23, line 573: What are these restrictions?

Author's response: The spatial limitations of T. vitticeps are attributed to its concentrated distribution within the southeastern region of the Atlantic Forest biome in the state of Espírito Santo. An explanation regarding these restrictions has been included in the revised version of the text on page 27, lines 649-654. 

- page 23, line 584-585: Example to use data from mapbiomas.

Author's response: We thank the Reviewer for suggesting the use of data available on the MapBiomas platform (which are also easily accessible through Google Earth Engine), however, the use of categorical variables together with continuous variables may result in models with low statistical quality. Thus, the MapBiomas variables were not used in this work.

“- page 24, line 608-609: Example to use the NDVI PCAs.”

Author's response: To carry out the analysis of the degree of contribution of the variables that contributed to the model, the method of Partition of Variance was used in this work, which allows identifying the percentage of importance of each variable for the construction of the model, as well as the residuals referring to the results in which none of the variables were able to explain, using the canonical redundancy analysis (RDA), as explained 272-289 on page 7.

- Put the caption of Figures 4 and 6 on the same scale (0 - 1).

Author's response: We corrected the maps in Figures 4 and 6 so that the scales remained standardized from 0-1.

“- I recommend placing the caption inside figure 9, as it was done in figure 10. This improves the quality of the images/figures.”

Author's response: Due to the size of the image and the number of color classes, it was not possible to adjust the legend to the interior of the map. However, we made adjustments to Figure 9, so that it was possible to improve the quality and visualization of its legend.

“- Tables S3, S4, S5 and S6 could be grouped into a single table identifying the relative columns for each variable/group of variables.”

Author's response: We have made the necessary revisions by consolidating the previous Tables S3, S4, S5, and S6 into a single table, which is now referred to as Table S3.

“- Put the classes in Tables S7 and S8 in English.”

Author's response: We made the necessary corrections in the translation of the text of Tables S7 and S8 into English. The previous Tables S7 and S8 were renamed as Tables S4 and S5.

“I also missed the elevation variable (derived from the SRTM), as the altitude effect in the Atlantic Rainforest can represent an obstacle to the existence of opossums/vectors/parasites.”

Author's response: We are grateful for the Reviewer's observation, however, the use of the SRTM elevation variable was not included to analyze the host/parasite/environment triad in the landscape approach due to the characteristics of D. aurita, being a nomad and generalist species. The inclusion and exclusion buffers allow the representation of the species' ability to move as a function of environmental characteristics, mainly in relation to the NDVI, reflecting on the health of the vegetation. The variation in altitude that could come to influence the understanding of the ecological niche of D. aurita, and even triatomines, would not necessarily represent a reality of geographic limitation, only the altitude at which the point of occurrence is present, and that occurs because of spatial resolution (~1km²) and cartographic generalization. It is a variable considered indirect for the distribution of species, in which the associated temperature is a determining factor (physiologically) for its distribution. In addition, the altitude variable has a high correlation with climate variables (mainly those related to temperature), and it is not necessary to include it in the modeling because its characteristics are being represented by the climate. The altitude of the study area does not constitute a barrier for the triad, not being a limiting factor for the distribution of the species (vector and mammal). In view of this, the SRTM elevation variable, despite being an interesting variable, was decided not to be used in this work.

Reviewer #2: We thank the Reviewer for all the meticulous care he/she took in reading the work, both in correcting errors and in conceptually and methodologically guiding the work. His/her revision was of great importance for the improvement of the work, and so that we made the necessary corrections, which is highlighted in the version "Revised Manuscript with Track Changes". Below are the specific comments:

“This is a modelling study about the ecological niches of multiple species involved in the transmission of T. cruzi in the Brazilian Atlantic forest biome. The methods are carefully thought and well described, with some specific exceptions that I detail below. The results are original and relevant, but their presentation and discussion can be significantly improved by a major revision of the text. As a reader, I found the text hard to read, which hinders the communication of the main results.

My main concern about the methods is the Ecoland approach. I do not think that its results add much to the whole interpretation of the study, at least not in the current version of the manuscript. The maps on figures 9 and 10 have too many different categories and colours, making it hard to summarise any clear message out of them. The main objective of the study (as stated in the abstract) is achieved by the other results and is properly covered in most of the Discussion, whilst the Ecoland approach is only mentioned in the very last sentences of the section. I suggest rewriting a stronger justification to maintain its results. Aggregating the model outputs by municipality can be extremely relevant to decision-makers, but given the high number of municipalities, it is hard to summarise it in a single map. I suggest keeping them only as supplementary files, as they already are in S7 and S8 tables.”

Author's response: The Ecoland method is of great importance for this work due to its ability to simplify ecological niche models, from a pixel-level scale to a spatial-geometry scale, that is, it generalized tens, hundreds and even thousands of pixels within a District into just a single value representative of that District. Its analysis was adequately explored in depth in the discussion, with no need to improve it without causing redundancies or even adding unnecessary analyzes to the text. Although we understand the Reviewer's suggestion to include Figures 9 and 10 as Supplementary data, we believe that their presence within the text would enhance the reader's ability to visualize the potential of this technique. By employing map algebra to analyze the agreements and disagreements among the generated predictive models in the geographic space, these figures demonstrate how the technique simplifies and optimizes the predictive distribution of models at the district level. In our study, we validated this approach in regions with cases and/or outbreaks of Chagas disease. The inclusion of these figures allows for the presentation of combined results from different models, representing pixel value aggregation on a scale of administrative political divisions. This facilitates the recognition of territories by decision makers, aiding in the formulation of public policy actions. Moreover, it helps identify target districts that require intensified efforts in terms of human and financial resources to surveillance and control T. cruzi transmission effectively. We appreciate the author's suggestion, but we have decided to retain Figures 9 and 10 in the main text.

“14-15: This sentence implies that the authors modelled the niche of the parasite, which does not seem to be the objective of the study.”

Author's response: We thank you for the suggestion, and we have corrected the text in lines 14-15 of page 1, present in the Abstract.

“72: Jansen et al. (missing ref)”

Author's response: Thank you for identifying the error that went unnoticed, and we have made the necessary corrections on line 77 on page 2.

“91: Be careful with the concept of model extrapolation (here and throughout the text). The general ENM approach aims at prediction within the ranges of the variables in the training set. Whether or not strict extrapolation occurs, it depends on the choice of algorithm and the whole modelling approach. See Qiao et al. 2018 (https://doi.org/10.1111/ecog.03986) for clarification.”

Author's response: As suggested by the Reviewer, we have adjusted the text, now on page 3, to lines 97-99. The word "extrapolated" was replaced to adjust the text according to the concepts and practices of the ENM, according to the text: "The ENM can estimate the Existing Fundamental Niche ("potential niche") of a species through the environmental characteristics where its points of occurrence are located, resulting in models that represent areas with adequate environmental characteristics for their maintenance [23,24]."

The same error that is repeated in other points of the text, as in lines 552 and 559 of page 23, were replaced by the word "predicted", and are now present in lines 620 and 628 of page 27.

“182: Please describe the criteria for establishing the radii for each species.”

Author's response: An explanation of the of radius for the inclusion and exclusion buffers of each species was included in lines 190 - 198, on page 5: "The T. vitticeps exclusion buffer radius was 10 km [34], while its inclusion buffer was 60 km, approximately 90% of the median of the matrix of distances between occurrences. The exclusion buffer for P. megistus was the same as for T. vitticeps, while the inclusion buffer, due to its greater environmental diversity because of its greater spatial distribution along the Atlantic Forest, was defined as being 40% more than median of the matrix of distances between occurrences of T. vitticeps. For D. aurita, because it is a nomadic species with great displacement capabilities and presence in anthropic areas [35,36], its exclusion buffer was defined as twice the buffer of T. vitticeps (20 km), while its inclusion buffer was 100 km, the same as P. megistus.".

“199: Please describe the method to assess the Pearson's correlation? (by using the full set or a sub-selection of pixel values)”

Author's response: The correlation index was calculated considering only the pixels present in the Atlantic Forest area, increased by 50 km, considering the random selection of 1000 pixels, and this explanation was included in the text, on lines 216 - 218 on pages 4-5.

“225-233: Consider rewriting this paragraph for a better description of the method. Were the occurrences artificially selected if they were associated with NDVI values above 0.2?”

Author's response: Yes, the points were artificially removed due to the characteristics of the occurrence points in the database in relation to the NDVI. This allowed reducing the redundancy of the environmental characteristics of the occurrence points without affecting the NDVI variable, since it has a drastic change in values between neighboring pixels. The text has been corrected as represented on lines 248 - 257, on page 6: "However, as D. aurita was modeled using NDVI in a landscape approach, to perform this redundancy correction of the environmental characteristics, the removal of nearby points was carried out , associating them with this variable, thus avoiding the loss of important information of the vegetation for the species, as the pixel values of this index can vary abruptly between neighboring pixels. For this, the occurrence points that were in intersection areas within the same radius of 1.5 km from each point would be removed according to the NDVI value of these points, removing all that were below 0.2, and/or keeping the point with the highest NDVI value in that area, assuming it is greater than 0.2. approximately 99% of the occurrence points of the D. aurita bank are contained above this value. For T. cruzi infection, there was no need to apply this filter, as the number of points was already low (29 points), and the points were already spaced apart (>1.5 km)."

“239: There is no mention of these acronyms elsewhere in the text. Explain what those models are.”

Author's response: Corrections were made to the text, referencing Table S2.

“239-241: Give more information about the variance partition procedure. Was this done over the final ensemble models? How to assess variable contribution in the ensemble models, after the predictions were averaged?”

Author's response: Yes, the Partition of Variance was performed as a function of the final Ensemble model. This method uses the canonical analysis by analysis of redundancy (RDA) considering the Ensemble model as a Y response variable and the environmental variables as the explanatory variables, obtaining the degree of contribution of each variable by fractions (Venn diagram groups) in relation to each model. The RDA uses eigenvectors, in a multiple linear regression, to verify the relationship of the matrix of response variables (Y) with the matrix of explanatory variables. The mathematical explanation for this method can be seen in the references: 38–41. The text has been adjusted on lines 272-288 on page 7: "After modeling M1, M2, M3, M4 and M5 (S2 Table), the Partition of Variance [38–41] was carried out to verify the degree of contribution (Aj R²) of each of the variables in the definition of the ecological niche for each of the Ensemble models generated, using the Venn Diagram to visualize the results. In the case of models generated by climate and landscape variables for D. aurita (M2 and M4), the components were divided into two groups of variables, temperature (BIO 2, BIO 4, BIO 5, BIO 8 and BIO 9) and of precipitation (BIO 12, BIO 13 and BIO 15), in the climate, while for the landscape each variable was presented as a component of the diagram (EDNHP, EDNDS and NDVI), that is, three components were generated. However, for the climate and landscape models for D. aurita infected by T. cruzi (M3 and M5), the Venn Diagrams were divided into 4 components in both cases, with the intention of allowing a better visualization and understanding of the temperature variables in the climate approach. For M3, two climatic components were produced (BIO 2, BIO 4 and BIO5) and (BIO 8 and BIO 9), a precipitation component (BIO 12, BIO 13 and BIO 15) and a component with the variable that represents the ecological niche models of triatomines (Triatomines). For M5, each variable was presented as a component of the diagram (EDNHP, EDNDS and NDVI), that is, three components were generated, and the fourth component was the variable that represents the ecological niche models of triatomines (Triatomines). This analysis was performed with R package vegan 2.6-4." However, the variance partition operating methodology, its mathematical essence, is being explained in references 38, 39, 40, 41 and 43, present in the method paragraphs.

“550 and 565: Review missing symbol font”

Author's response: We perform the adjustment on line 106, page 3. We have included in the text the meaning of the Beta symbol, which in this work represents Environmental Suitability (β).

“552-553: If model extrapolation was not specifically assessed, this could be the correct prediction from the models.”

Author's response: We thank you for identifying the conceptual error present in the text, and the word “extrapolated” were replaced by the word "predicted” and are now present in lines 620 and 628 of page 27.

“559: Same as before, maybe change “extrapolated” to “predicted””

Author's response: We thank you for identifying the conceptual error present in the text, and the word “extrapolated” were replaced by the word "predicted” and are now present in lines 620 and 628 of page 27.

“598-600: This sentence would be better suited in the Results section.”

Author's response: We adjusted the paragraph to fit the Discussion, correcting the text on lines 690-695 on page 28: "The temperature variables (BIO2, BIO4, BIO5, BIO8 and BIO 9) were the ones that most impacted the modeling of D. aurita, while precipitation alone contributed with only 3% for the modeling (S1 Fig), and this is a scenario that is related to the characteristics of the species, since the temperature is one of the factors that influence the beginning of the reproduction season of the year, while the Precipitation, while not having a large impact on the model, does influence food availability [54]."

“702-710: Rewrite for clarity.”

Author's response: The text was adjusted, on lines 805-813, on pages 30 and 31, excluding the phrase "As an example, precipitation is a factor that hinders its flight and makes the contact surfaces less adherent, influencing the displacement of the triatomine.", which was causing difficulty in understanding the paragraph. The final text became: "Of the variables used to model the ecological niche of D. aurita with positive blood culture, the temperature variables prevailed in the explanation of the model (Fig 8A). Superficially – and simply – this information is leading to the idea that the presence of positive blood culture in marsupials depends on the life dynamics of triatomines as a function of climatic relationships in space. However, the parasite is deeply involved in a trophic network, and the opossum does not necessarily depend on vector-contamination routes and/or predation by triatomines to become infected and may acquire the infection through the food web of infected small mammals."

Fig 1: "DB" is mentioned in the legend, but it is not in the figure.

Author's response: We removed "DB" from the text because it is not cited elsewhere and is not cited in Figure 1.

“Fig 5: What does the vertical red line represents? Please describe it in the legend.”

Author's response: The red line represents the cut-off value for the TSS of partitions with good statistical quality, and we made the adjustments in the captions of Figures 5 and 7, including this information. 

“Fig 8: This figure is hard to interpret, given the lack of details about the partition of variance method. What does the values in the intersection between variables mean? What is the difference between an empty intersection and the 0.00 value?”

Author's response: The values present in the intersection areas represent the degree of joint contribution of certain environmental variables to the Ensemble model. Values < 0 are interpreted as zeros, corresponding to cases where explanatory variables explain less variation than normal random variables, and values equal to zero, which are represented, mean that the degree of contribution between the set of variables is null for the variable response, which in this case is the Ensemble model. We improved the text on lines 510 - 511 on page 22, and on lines 971 - 972 on page 34, in the legends for Figures 8 and Fig S1, so that this question could be clarified.

“Figs 9 and 10: Too many categories/colours make it hard to summarise or extract any spatial trends from these figures”

Author's response: Figure 9 was readjusted so that the caption could have a more adequate visualization. However, the purpose of Figures 9 and 10 is not to make the reader identify in detail the characteristics of the presence of T. cruzi in D. aurita in your District (although Tables S4 and S5 have this information), but rather to make the reader visualize the potential that the Ecoland approach must analyze ecological niche models, making them simplified. Therefore, the interpretation of the map is complex, but of fundamental importance for the work.

“Supplementary files: Consider adding titles and legends to all tables inside the files.”

Author's response: We made the corrections and included the captions within each supplementary table.

“S2 Table: Did the climatic models (M2) include NDVI and the environmental distances? This would be contradictory with S1 Fig and some parts of the text.”

Author's response: No, they don't. This was a typo, which has been corrected in Table S2, where the NDVI and distance variables are only present in the landscape approaches of D. aurita and D. aurita infected by T. cruzi, being M4 and M5 respectively. The correction was made to Table S2 by removing the landscape variables that were incorrectly present in M2.

Reviewer #3: We would like to thank the Reviewer for carefully reading the text, suggesting improvements and highlighting the importance of the work. All corrections suggested by the Reviewer were considered and adjusted in the text. All line and page references are to the marked version of the text “Revised Manuscript with Track Changes”. Here is the answer for the specific comments:

This is an interesting and well written manuscript. This work is well addressed and the subject is clearly relevant to the journal. The data collected represent a very impressive and interesting dataset, the models are compelling and the results are original and clearly. The outcomes could contribute to a better understanding of T. cruzi transmission in disturbed environments, where generalist reservoirs such as D. aurita are abundant (see attachment for comments).

Manuscript title: Space-environment relationship in the identification of potential areas of expansion of Trypanosoma cruzi infection in Didelphis aurita in the Atlantic Rainforest

This is an interesting and well written manuscript, which combines information on the ecology and epidemiology of Trypanosoma cruzi in the Atlantic rainforest, with the use of valuable tools such as Ecological Niche Modeling and the Ecoland approach. This work is well addressed and the models are compelling. 

The broad objective of this study was to understand the conditions related to the parasite/host/environment triad in which the transmission of T. cruzi occurs involving a generalist wild species (strongly associated with anthropic changes) such as Didelphis aurita in the Atlantic Rainforest, a biome whose coverage area has been reduced in recent years, and currently, only 12,4% of its original area is well conserved. The authors hypothesized that the spatial distribution of triatomines Triatoma vitticeps and Panstrongylus megistus, the host D. aurita and their infection by T. cruzi is directly related to the physiognomy of the landscape. 

Applying ecological niche models, the authors explored the distribution of infection by D. aurita and T. cruzi, using abiotic and biotic variables, predicting their occurrence. Records of T. cruzi infected and non-infected D. aurita (detected by hemoculture) were analyzed through climate and landscape approaches by the Ecoland method. The subject is clearly relevant to the journal, the data collected by the authors represent a very impressive and interesting dataset. The results are original and clearly within the scope of the journal. The outcomes could contribute to a better understanding of T. cruzi transmission in disturbed environments, where generalist reservoirs such as D. aurita are abundant. 

I have a concern regarding the use of blood cultures as the only parasitological technique, to determine positivity for T. cruzi, without considering the parasite load and infectivity to the vector. I understand that given the time range considered in this work (1992-2019), there are limitations regarding the availability of comparable results and techniques obtained from different databases.

Author’s response: The parasitological database used in the work comes from a single laboratory, which also harbors an Institutional Trypanosoma Collection of Wild, Domestic and Vector Mammals, COLTRYP/Fiocruz, as specified in the methodology, and that blood culture was chosen precisely because it is the technique performed by the same group of researchers and always under the same conditions during field expeditions. These characteristics allow the results to be fully comparable. The positive blood culture, in addition to being the gold standard, detects the parasite and certifies the infective competence, and in the parasitological database used, originating from only one laboratory, the parasites detected in all positive blood cultures were taxonomically characterized by molecular analysis.

However, given that the high reservoir competence of Didelphis genus is known, an interesting point to evaluate is the parasite load in different spatial and environmental contexts, including this variable in the models. Although I understand that this question is outside the scope of this manuscript, in a scenario where environmental changes prevail, there are multiple variables associated with sylvatic cycles and T. cruzi transmission, and the results obtained lead me to wonder about this point. In this study the models showed that areas of high suitability for the presence of marsupials are a necessary, but not sufficient, condition for D. aurita to act as a reservoir for T. cruzi. I encourage authors to explore, perhaps on a smaller scale, the parasite loads in blood of D. aurita as a better approximation to the reservoir competence of this species.

Author's response: Hemoculture is indicative of a higher parasite load, indicating the potential for transmission to the vector, and this clearly differentiates animals with or without infective potential. As for the comparison of parasite load between animals with positive blood cultures, this is not possible because cultures are an indirect parasitological test, which depend on growth in culture medium, and different parasite populations have different growth rates that depend on characteristics of each isolate, initial inoculum load and growth adaptation in axenic medium.

In a changing environmental context, the conclusions of this work are an important management tool in integrated strategies under the One Health approach.

Minor points

“Line 64-69: Although hemoculture and blood fresh examination are used, I suggest including xenodiagnosis as a very useful parasitological technique to determine the reservoir competence of sylvatic hosts of T. cruzi.”

Author's response: The correction was made, including the xenodiagnosis test on line 72, page 2.

“Line 819: delete the word funding”

Author's response: Correction performed on line 935 - 944, page 33-34, removing the Financing topic from the text.

“Consistent use of abbreviations. For example: use ENM to abbreviate Ecological Niche Modeling, throughout the manuscript (sometimes it appears abbreviated and other times complete).”

Author's response: Correction performed on lines 95 – 115, page 3.

We again acknowledge the improvements suggested by the reviewers and we are certain that the review process was essential to improve the quality of our data. 

Best wishes,

Corresponding author: Samanta Cristina das Chagas Xavier 

E-mail: sam.azeredo@gmail.com

---

## [Decision Letter · Decision Letter 1]

29 Jun 2023

Space-environment relationship in the identification of potential areas of expansion of *Trypanosoma cruzi* infection in *Didelphis aurita* in the Atlantic Rainforest

PONE-D-23-05941R1

Dear Dr. Xavier,

We’re pleased to inform you that your manuscript has been judged scientifically suitable for publication and will be formally accepted for publication once it meets all outstanding technical requirements.

Kind regards,

Everton Falcão de Oliveira, Ph.D

Academic Editor

PLOS ONE

Additional Editor Comments (optional):

Reviewers' comments:

Reviewer's Responses to Questions

**Comments to the Author**

1. If the authors have adequately addressed your comments raised in a previous round of review and you feel that this manuscript is now acceptable for publication, you may indicate that here to bypass the “Comments to the Author” section, enter your conflict of interest statement in the “Confidential to Editor” section, and submit your "Accept" recommendation.

Reviewer #1: All comments have been addressed

2. Is the manuscript technically sound, and do the data support the conclusions?

Reviewer #1: Yes

3. Has the statistical analysis been performed appropriately and rigorously? 

Reviewer #1: Yes

4. Have the authors made all data underlying the findings in their manuscript fully available?

Reviewer #1: Yes

5. Is the manuscript presented in an intelligible fashion and written in standard English?

Reviewer #1: Yes

6. Review Comments to the Author

Reviewer #1: Thank you for submitting your study to "Plos ONE".

All comments have been addressed. The only one that wasn't realized was a suggestion (about PCA).

7. PLOS authors have the option to publish the peer review history of their article (what does this mean?). If published, this will include your full peer review and any attached files.

Reviewer #1: No

---

## [Editor Report · Acceptance letter]

20 Jul 2023

PONE-D-23-05941R1 

Space-environment relationship in the identification of potential areas of expansion of Trypanosoma cruzi infection in Didelphis aurita in the Atlantic Rainforest 

Dear Dr. Xavier:

I'm pleased to inform you that your manuscript has been deemed suitable for publication in PLOS ONE. Congratulations! Your manuscript is now with our production department. 

Kind regards, 

on behalf of

Dr. Everton Falcão de Oliveira 

Academic Editor

PLOS ONE